# BET Bromodomain Inhibitors: Novel Design Strategies and Therapeutic Applications

**DOI:** 10.3390/molecules28073043

**Published:** 2023-03-29

**Authors:** Kenneth K. W. To, Enming Xing, Ross C. Larue, Pui-Kai Li

**Affiliations:** 1School of Pharmacy, Faculty of Medicine, The Chinese University of Hong Kong, Hong Kong, China; 2Division of Medicinal Chemistry and Pharmacognosy, College of Pharmacy, The Ohio State University, Columbus, OH 43210, USA; 3Department of Cancer Biology and Genetics, College of Medicine, and Division of Pharmaceutics and Pharmacology, College of Pharmacy, The Ohio State University, Columbus, OH 43210, USA

**Keywords:** BET inhibitors, bromodomain, Brd4, epigenetics, extra-terminal domain, PROTACs

## Abstract

The mammalian bromodomain and extra-terminal domain (BET) family of proteins consists of four conserved members (Brd2, Brd3, Brd4, and Brdt) that regulate numerous cancer-related and immunity-associated genes. They are epigenetic readers of histone acetylation with broad specificity. BET proteins are linked to cancer progression due to their interaction with numerous cellular proteins including chromatin-modifying factors, transcription factors, and histone modification enzymes. The spectacular growth in the clinical development of small-molecule BET inhibitors underscores the interest and importance of this protein family as an anticancer target. Current approaches targeting BET proteins for cancer therapy rely on acetylation mimics to block the bromodomains from binding chromatin. However, bromodomain-targeted agents are suffering from dose-limiting toxicities because of their effects on other bromodomain-containing proteins. In this review, we provided an updated summary about the evolution of small-molecule BET inhibitors. The design of bivalent BET inhibitors, kinase and BET dual inhibitors, BET protein proteolysis-targeting chimeras (PROTACs), and Brd4-selective inhibitors are discussed. The novel strategy of targeting the unique C-terminal extra-terminal (ET) domain of BET proteins and its therapeutic significance will also be highlighted. Apart from single agent treatment alone, BET inhibitors have also been combined with other chemotherapeutic modalities for cancer treatment demonstrating favorable clinical outcomes. The investigation of specific biomarkers for predicting the efficacy and resistance of BET inhibitors is needed to fully realize their therapeutic potential in the clinical setting.

## 1. Introduction 

In eukaryotic cells, DNA is packed in a highly organized chromatin structure within the nucleus by wrapping around histone proteins. The packaging of DNA plays a critical role in controlling the interaction of various regulatory proteins with the *cis* elements in the promoters of genes. Post-translational modifications (PTMs) of histone proteins, including acetylation, methylation, phosphorylation, ubiquitylation, SUMOylation, glycosylation, and ADP-ribosylation, are known to shape the chromatin configuration and function [1]. These histone PTMs are commonly known as epigenetic marks, which are critical in regulating chromosome packaging, gene transcription, mitosis, meiosis, apoptosis, and DNA damage. Defects in these PTMs are associated with cancer progression [2]. 

The acetylation of the lysine tails of histone proteins is considered the most dynamic PTM, which controls chromatin configuration and its interaction with other proteins [3]. As histone lysine acetylation is generally associated with transcription activation, dysregulation of histone acetylation usually leads to the aberrant expression of oncogenes and subsequently promotes cancer development [4]. There are three types of proteins regulating histone acetylation, which include bromodomain (BRD) proteins [5,6], histone acetyltransferases [7], histone deacetylases [8], and sirtuins [9]. Acetyltransferases, BRDs, and deacetylases and sirtuins, are considered the “writers”, “readers”, and “erasers” of histone acetylation, respectively [10]. 

BRDs are evolutionally conserved protein–protein interaction motifs that recognize acetylated lysine residues on histones, thus playing a critical role in chromatin remodeling [6]. They recruit other proteins to transcriptional active sites of various genes and regulate the activities of chromatin remodeling complexes to control gene expression [11]. 

The bromodomain and extra-terminal domain (BET) family of proteins is the most extensively studied cellular protein carrying the bromodomains. They are linked to cancer progression due to their interaction with numerous cellular proteins including chromatin-modifying factors, transcription factors, and histone modification enzymes. Current approaches targeting BET proteins for cancer therapy rely on acetylation mimics to block the interaction between bromodomains and chromatin. However, bromodomain-targeted agents are suffering from dose-limiting toxicities because of their effects on other bromodomain-containing cellular proteins. This review aims to provide an updated summary about the evolution of small-molecule inhibitors of BET proteins. The design of bivalent BET inhibitors, kinase and BET dual inhibitors, BET protein proteolysis-targeting chimeras (PROTACs), and Brd4-selective inhibitors will be discussed. The novel strategy of targeting the unique C-terminal extra-terminal (ET) domain of BET proteins and its therapeutic significance will also be highlighted. Apart from single agent treatment alone, BET inhibitors have also been combined with other chemotherapeutic modalities for cancer treatment demonstrating favorable clinical outcomes.

## 2. BET Family of Bromodomain Proteins Represent Attractive Targets for Cancer Therapy 

### 2.1. Bromodomains (BRDs)

“Readers” of the various epigenetic marks are structurally diverse proteins, which harbor evolutionarily conserved motifs to recognize the covalent modifications of histone proteins or DNA [10]. BRDs are well-known “readers” of the histone acetylation marks [12]. They are structurally conserved and consist of about 110–120 amino acid residues. They are present in many chromatin-associated factors, including histone acetyltransferases (HATs), chromatin remodeling factors, and the BET family of proteins. To date, there are 61 bromodomains encoded in the human genome, which can be categorized into eight subfamilies according to their sequence and structural similarity [12] (Table 1). The aberrant regulation of numerous BRD-containing proteins is implicated in the etiology of various disease processes such as cancer, inflammation, and viral replication [13,14,15,16]. With the intensive development of small-molecule inhibitors of BRD proteins in recent years, results from the chemical-biology-based investigation have provided an in-depth understanding of the role of BRD proteins in numerous biological processes and ascertained BRD proteins as genuine drug targets in various diseases [12]. 

### 2.2. Bromodomain and Extra-Terminal Domain (BET) Proteins

BET proteins possess two N-terminal bromodomains (BD1 and BD2), a common structural feature with other BRD proteins, that allow them to interact with acetylated lysine residues on histone [6]. On the other hand, BET proteins are structurally distinct from other bromodomain-containing proteins in that they contain a C-terminal extra-terminal (ET) domain [17]. This ET domain interacts with a variety of cellular proteins such as histone-lysine N-methyltransferase (NSD3) [18,19,20,21] and Jumonji domain-containing 6 (JMJD6) [22,23], whose interactions are implicated in acute myeloid leukemia (AML) and various solid tumors, respectively. The combination of dual BRDs and the protein-interacting ET domain allows BET proteins to effectively mediate the bimodal tethering of these cancer-associated factors to specific regions of chromatin.

There are four mammalian conserved members of BET proteins, which include BRD containing 2 (Brd2), Brd3, Brd4, and Brdt [5]. While Brdt is predominantly expressed in germ cells, Brd2, 3, and 4 are ubiquitously expressed in various tissues [5]. All BET proteins adopt a left-handed four-helix bundle structure (α_Z_, α_A_, α_B_, and α_C_), called the “BRD fold” [24]. The inter-helical α_Z_-α_A_ (ZA) and α_B_-α_C_ (BC) loops constitute a hydrophobic pocket that recognizes the acetylated lysine residues [24]. The sequence variations in the ZA and BC loops of the distinct BRD fold in different BET proteins lead to their differential protein-binding sites and affinities [25]. It has been proposed that multiple BET proteins are required for the rapid induction of selected target genes [26]. Thus, different BET proteins may have non-overlapping functions, and they may form protein complexes with one another to elicit their biological activities. 

BET proteins are transcription regulators. Brd4 is known to recruit PTEF-b (positive transcription elongation factor which is a multi-protein complex essential for transcription regulation) to sites of active transcription of cell growth-promoting genes such as *MYC* and *NUT* [27,28]. During transcription elongation, Brd4 directs the proper nuclear localization and activation of PTEF-b to phosphorylate RNA polymerase II [29]. Therefore, Brd4 plays a critical role in facilitating the enhancement of basal transcription to active elongation by RNA polymerase II. Meanwhile, the ET domain of Brd4 is known to recruit other transcriptional activators including NSD3, JMJD6, and CHD4 to further promote gene transcription [18]. On the other hand, Brd3 binds specifically to the GATA1 transcription factor and upregulates the expression of GATA1-dependent genes [30]. Brd2 is known to interact with E2F, histone acetyltransferases, and histone deacetylases and recruit them to gene promoters, thereby coupling histone acetylation to transcription in a PTEF-b-independent manner [28]. 

The BET family of proteins (Brd2, 3, 4, and T) are of significant clinical interest due to their role in cell cycle regulation, epigenetic sensing, and a range of cancers from oral, breast, prostate, lung, colon, to myeloid leukemia. BET proteins contain dual bromodomains, a domain present in a variety of cellular proteins, which selectively bind acetylated histone marks. These include histone-lysine N-methyltransferase protein ASH1L, histone acetyltransferase p300 (EP300), P300/CBP associated factor (PCAF), and the extended BET family. Current therapeutic approaches rely on small-molecule acetylation mimics which block the ability of the bromodomains to bind their specific chromatin marks. Cancer inhibitors targeting bromodomains have seen limitations in the clinic due to dose-limiting toxicities as they target any proteins containing bromodomains.

### 2.3. Brd4

Brd4 is the most extensively studied member of the BET protein family. It is involved in numerous normal cellular functions and the development of different diseases such as cancer, cardiovascular diseases, chronic inflammation, and viral infection [31,32,33]. In normal healthy cells, Brd4 is required for the maintenance of chromatin stability, and it also controls cell cycle progression from the M phase to the G1 phase, through its interaction with PTEF-b and its role as a global transcriptional coactivator [34]. Animal studies showed that heterozygous Brd4+/− mice suffered from severe defects in cell differentiation and organogenesis [28]. Moreover, Brd4-null mouse embryos were reported to die in utero because of their inability to sustain the development of embryonic stem cells [28]. 

Brd4 has attracted enormous attention as a promising molecular target for cancer therapy in recent years because pharmacological inactivation of Brd4 was shown to remarkably suppress expression of the *MYC* oncogene [35]. While MYC is a well-known oncogenic transcription factor and a major driver for a wide variety of cancers, no effective drugs that directly target MYC are currently available. Thus, the pharmacological inhibition of BRD4 represents a promising indirect anticancer strategy to inhibit the hyperactive MYC in tumors. 

### 2.4. Brd4 at Super-Enhancers of Actively Transcribed DNA

In the human genome, enhancers are short (50–1500 bp) non-coding sequences that are bound by various activators, and they cooperate with gene promoters to enhance target gene transcription. In regions of actively transcribed DNA, super-enhancers are formed by clusters of enhancers that are densely packed with Brd4, acetylated histones, transcription factors, Mediator, and other coactivators [36]. Super-enhancers can span as long as 50 kb pairs, and they are located in close proximity to pre-selected gene promoter regions to prime the genes for strong and rapid activation [37]. At oncogenic super-enhancers, elevated levels of histone acetylation (particularly H3K27ac), Brd4, and Mediator facilitate chromatin remodeling and aggregation of transcriptional machinery to promote gene transcription [38]. The close proximity of super-enhancers to various oncogenes, including *MYC*, *IGLL5*, *IRF4*, and *XBP1*, to drive oncogene overexpression has been extensively studied [39]. A schematic diagram showing the indispensable role of Brd4 in assembling a super-enhancer is depicted in Figure 1. Importantly, the loss of Brd4 function or availability was shown to cause the preferential loss of other core transcriptional co-regulators from super-enhancers, subsequently suppressing the overexpression of oncogenes [40]. There, observations underscore the central role played by Brd4 in super-enhancer-driven oncogene activation. 

### 2.5. BRD4–NUT Fusion 

NUT midline carcinoma (NMC) is probably the first and most well-documented cancer type that is driven by genetic abnormalities associated with the BET family of proteins. It is a rare and aggressive subset of squamous cell carcinoma, which is characterized by the translocation and fusion of the amino terminus of *BRD4* (or less commonly *BRD3*) to the carboxyl terminus of *NUT* [41]. The resulting fusion oncoprotein Brd4-–NUT retains the two bromodomains and the ET domain, thus allowing it to bind with numerous transcriptional coactivators (particularly p300) [42]. It creates a positive feedback loop that subsequently generates “megadomains” of hyper-acetylated and active chromatin [43]. Interestingly, despite the abnormal size of the megadomains, only a specific subset of genes is aberrantly upregulated by the Brd4–NUT fusion (most notably MYC and TP63) [43]. The median survival of NMC is only about 6.7 months [44]. NUT tumors are usually found in the head, neck, and mediastinum, but they can also occur in the adrenal gland, bladder, kidney, and pancreas [45]. As the bromodomains of Brd4–NUT are driving the oncogenic function of the fusion protein, there is strong rationale to evaluate the potential treatment of NMC with BET inhibitors. 

With the increased understanding of tumor types that are regulated by BET/Brd4-associated oncogenes [46,47] and those driven by the Brd4 fusion [48,49], there has been enormous interest in developing novel BET inhibitors for cancer treatment. A number of the novel candidates are currently progressing through early clinical trials. The following section describes the design strategies and progress of the various types of BET inhibitors in preclinical and clinical studies. 

## 3. Evolution of Small-Molecule Inhibitors of BET Proteins

### 3.1. Pan BET Inhibitors 

#### 3.1.1. JQ1 and Its Analogs

Filippakopoulos et al. were the first to report a cell-permeable small-molecule inhibitor (JQ1) of BET proteins [49]. JQ1 (thieno-triazolo-1,4-diazepine) (Figure 2A) was designed to mimic acetylated lysine. It binds competitively and specifically to the BD1 and BD2 bromodomains with high affinity, forming a hydrogen bond with a conserved asparagine residue at the binding pocket [49]. Using chromatin immunoprecipitation and fluorescence recovery after photobleaching assays, JQ1 was shown to displace Brd4 from chromatin and thus modulate bromodomain-regulated genes [49]. JQ1 exhibits a potent antiproliferative effect in Brd4-dependent cancer cells lines and against NMC [50]. The work paved the way for the further development of other BET inhibitors, currently being actively investigated clinically for cancer therapy. 

However, JQ1 exhibits poor oral bioavailability, and it has a short half-life of only 1 h [51]. TEN-010 (also known as JQ2) (Figure 2B) is an analog derived from JQ1 [52]. It is currently investigated in clinical trials for patients bearing AML, myelodysplastic syndrome (MDS), and a few other solid cancers (NCT01987362, NCT02308761). Initial clinical data suggest that TEN-010 possesses more favorable pharmacokinetic properties than JQ1, including higher t_max_ and longer half-life [53]. OTX015 (Figure 2C) is another pan-BET inhibitor structurally similar to JQ1, with improved oral bioavailability [54]. It was shown to inhibit BRD2, BRD3, and BRD4. OTX015 demonstrated good anticancer activity in hematological cancers (B-cell lymphoma and multiple myeloma) and some solid cancer types (mesothelioma and neuroblastoma) [51,55,56]. 

#### 3.1.2. I-BET762 (GSK525762A, Molibresib) 

I-BET762 (Figure 2D), developed by GlaxoSmithKline, is an orally administrated pan-BET inhibitor with activity against BRD2, BRD3, and BRD4 [57,58]. It was initially designed as a novel drug candidate to reduce inflammation by dissociating the BET family of proteins from the enhancer region of inflammatory genes in activated macrophages [57]. After the initial success of BET inhibition by I-BET762, it was also shown to exhibit potent anticancer activities in various preclinical cancer models, including neuroblastoma and pancreatic cancer [59,60]. Recently, I-BET762 was also investigated in a clinical trial for the treatment of NMC, small-cell lung cancer, castration-resistant prostate cancer, triple-negative breast cancer, and gastrointestinal stromal cancer [61]. However, the adverse effects observed for I-BET762 were similar to those reported for OTX015, suggesting it may share its toxicity profile with other earlier-generation BET inhibitors [61]. 

#### 3.1.3. BET-PROTACs

Proteolysis-targeting chimeras (PROTACs) are chimeric bifunctional small molecules that recruit an E3 ligase to promote the degradation of a target protein [62]. Structurally, a typical PROTAC is composed of a ligand capable of binding to an E3 ligase and another ligand that binds to the target protein through a linker. Therefore, PROTACs facilitate the formation of a trimeric complex to promote ubiquitination and subsequent degradation of the therapeutic protein target by the proteasome. Figure 3 depicts the general design format of a BET-targeting PROTAC. Compared with conventional inhibitors, PROTACs not only bind to the molecular targets but also possess the unique ability to eliminate protein targets that are not “druggable” by traditional small-molecule inhibitors [63] or those that are non-enzymatic proteins (e.g., transcription factors) [64,65]. There has been extensive research in recent years to design PROTACs for various therapeutic applications. Two comprehensive reviews have been recently published about the application of BET-protein-targeting PROTACs for cancer therapy [66,67]. 

Early-generation BET-targeting PROTACs exploit cereblon (CRBN) E3 ligase in their design [64]. CRBN is the substrate receptor within the CRL4^CRBN^ E3 ubiquitin ligase, which is the primary target of thalidomide and its analogs [68]. By directly binding to CRBN, the thalidomide analogs of the PROTAC system serve as a ligand to facilitate the formation of a ternary complex between CRL4^CRBN^ and the target protein, subsequently leading to protein ubiquitination and proteasome-based degradation [69]. dBET1 (Figure 4A) is one of the first reported CRBN-based small-molecule PROTACs targeting the BET protein [70]. The molecule was designed by coupling (+)-JQ1 to the CRBN-recruiting ligand thalidomide via a flexible linker [71]. dBET1 exhibited high selectivity to BET protein over other bromodomains. Upon 2 h incubation of 100 nM dBET1 in an AML cell line (MV4–11), more than 95% of BRD4 was degraded, which was accompanied by a remarkable depletion of its target protein c-Myc. More recently, QCA570 (Figure 4B) was designed as an exceptionally potent CRBN-based and BET-targeting PROTAC [65]. Using lenalidomide as the CRBN ligand, QCA570 was shown to induce the specific degradation of Brd3/4 at a concentration as low as 10 pM in AML cell lines after 3 h of treatment. In various AML models, QCA570 achieved cancer cell growth inhibition IC_50_ of 8, 3, 62, and 32 pM in MV4–11, MOLM-13, and RS4–11 cells, respectively, which was 10- to 1000-fold more potent than dBET1 [65]. Importantly, QCA570 also produced a complete and durable tumor suppressive effect in an RS4–11 tumor xenograft model at an IV dose as low as 1 mg/kg three times a week, without appreciable toxic effects [65]. 

The therapeutic efficacy of BET-targeting PROTACs is often hindered by poor cell permeability, metabolic instability, unfavorable in vivo pharmacokinetics and in vivo efficacy, and undesirable toxicity due to off-tissue on-target protein degradation [72]. To address these problems, novel BET-targeting PROTACs have been designed using new strategies, including in-cell self-assembly, light-activated, aptamer PROTAC conjugation, antibody-coupling, and autophagy-targeting strategies [67]. 

JQ1-CLIPTAC (Figure 4C) was a novel Brd4-targeting PROTAC designed with an aim to improve the unfavorable cellular permeability of PROTACs [73]. Interestingly, JQ1-CLIPTAC was generated within the cells by a self-assembly Diels–Alder click reaction between a *trans*-cyclooctene-labeled (+)-JQ1 (JQ1-TCO) (Figure 4D) and a tetrazine-labeled thalidomide Tz-thalidomide [73]. Cancer cells were treated with Tz-thalidomide (10 μM) for 18 h followed by (+)-JQ1-TCO (3 μM) for another 18 h, or *vice versa*, to give rise to complete Brd4 degradation. In contrast, Brd4 protein expression was not altered if cancer cells were treated with “pre-prepared” (+)-JQ1-CLIPTAC (10 μM) due to the poor cellular permeability of the PROTAC [73]. 

In general, BET-targeting PROTACs are not tissue-specific. As the E3 ligases are mostly ubiquitously expressed in both tumor and normal tissues, the development of tissue-specific E3 ligases or new tumor-targeting delivery strategies will be needed to eliminate the off-tissue on-target BET degradation effect and unwanted toxicity [74]. To this end, BET degrader–antibody conjugates have been designed to improve the pharmacokinetic properties of PROTACs, facilitate antigen-based targeted delivery, and potentiate in vivo antitumor activity. Ab-PROTAC3 is a BET-degrading antibody conjugate recently reported by Tate et al. [75]. It consists of the humanized anti-HER2 monoclonal antibody (trastuzumab) linked to the BET degrader PRO via a cleavable ester linker with excellent stability in human serum. A detailed live-cell confocal microscopic study in HER2+ breast cancer cell model confirmed that Ab-PROTAC3 could be readily taken up by the cells via HER2-dependent endosomal internalization [75]. Once inside the cells, the BET degrader PRO was released and triggered BRD4 depletion following lysosomal trafficking. 

The aptamer–PROTAC conjugation is another approach to achieve target tissue-specific BET degradation. Aptamers are short, synthetic, single-stranded DNA or RNA oligonucleotides that can specifically bind to their target proteins, nucleic acids, or tissues. APR (Figure 4E) is the first aptamer-conjugated PROTAC reported [76]. It was designed by connecting the tumor-targeting nucleolin apatmer AS1411 to the BET PROTAC (PRO) via a glutathione-labile ester–disulfide linkage. The specific recognition of nucleolin by AS1411 allows the specific targeting of the PROTAC to cancer cells overexpressing nucleolin. Upon cellular uptake, APR is cleaved by glutathione at the disulfide linker to release PRO for degrading the BET protein. APR exhibited robust Brd4 degradation (with a EC_50_ = 22 nM), potent cancer cell growth inhibition (IC_50_ = 56.9 nM in MCF breast cancer cells in vitro), and pronounced antitumor activity in MCF-7 tumor xenografts in vivo (TGI = 77.5%). More importantly, APR did not cause notable tissue damage to any major organs, whereas the PROTAC alone produced severe lesions in lung tissues [76]. 

Light activation also represents a highly site-specific drug delivery approach for cancer therapy. Xue et al. was the first to report a light-controlled Brd4-targeting PROTAC (pc-PRORAC1) (Figure 4F) [77]. PRORAC1 was designed by including a photocleavable linker (4,5-dimethoxy-2-nitrobenzyl (DMNB) group) in dBET1. In Ramos cells without light irradiation, pc-PRORAC1 could not degrade Brd4 at a concentration as high as 3 μM. However, upon irradiation at 365 nm for 3 min, the DMNB group was removed from pc-PRORAC1 and generated dBET1 to effectively degrade Brd4. Under light conditions, pc-PRORAC1 exhibited a potent anticancer effect in Burkitt lymphoma Namalwa cells (GI_50_ = 0.4 μM), which was in the same range as that achieved by dBET1 (GI_50_ = 0.34 μM) [77].

The conventional BET-targeting PROTACs exploited the ubiquitin–proteasome system to degrade BET proteins. Instead, AUTAC-1 (Figure 4G) is a novel BET degrader that utilizes the autophagy pathway for BET protein degradation [78]. It was designed by tethering the LC3 ligand GW5074) to the BET inhibitor (+)-JQ1 via a PEG linker. LC3 is a characteristic autophagy protein located on the membrane surface of autophagosomes. Thus, AUTAC-1 could direct the delivery of the BET protein to the autophagosomes, subsequently leading to protein degradation via the autophagy lysosomal pathway [78]. 

It is noteworthy that acquired resistance to both CRBN- and VHL-based BET-targeting PROTACs has been reported in cancer cells after prolonged exposure [79]. It was found that the acquired resistance was associated with genomic alterations that compromise the core components of the CRBN or VHL E3 ligase complexes [79]. New BET-targeting PROTACs that seize novel E3 ligases (such as KEAP1, DCAF15, RNF114, and FEM1B) using covalent ligands have been reported recently to address the acquired drug resistance problem [67,80]. However, the potency and selectivity for their BET degradation still need to be optimized.

### 3.2. BD1- or BD2-Selective BET Inhibitors 

All of the four BET proteins have two conserved bromodomains (BD1 and BD2) that preferentially bind to peptide sequences bearing multiple acetylated sites [6]. The preferred ligand of BD1 is K^ac^XXK^ac^, where the intervening amino acids X usually have small side chains (e.g., alanine or glycine). On the other hand, BD2 is known to bind to acetylated sites within more diverse peptide sequences [6,81]. The early-generation pan-BET inhibitors exhibit equal affinity to both the BD1 and BD2 bromodomains of BET proteins. Extensive research has been conducted to design novel BET inhibitors with greater selectivity and less adverse effects. One approach is to target only one of the two bromodomains (i.e., either BD1 or BD2) of BET proteins [82]. Interestingly, most selective BET inhibitors developed were reported to target the BD2 domain. However, emerging evidence suggests that targeting the BD1 bromodomain may be sufficient to elicit anticancer activity [26]. On the other hand, BD2 was found to be more critical for the activation of interferon-response genes [82]. A recent study systematically investigated a panel of 18 BET inhibitors for their selectivity against BD1 versus BD2 bromodomains [83]. Two novel compounds, namely, GSK778 (iBET-BD1; BD1-selective) (Figure 5A) and GSK046 (iBET-BD2; BD2-selective) (Figure 5B), were designed by a structure-based algorithm to interact specifically to the BD1 and BD2, respectively, of BET proteins [26]. By surface plasmon resonance binding assay, GSK778 is > 130-fold selective for BD1, whereas GSK046 is > 300-fold selective for BD2 [26].

ABBV-744 (Figure 5C) is another newly developed BD2-selective BET inhibitor [84]. It exhibits a several-hundred-fold higher affinity for BD2 over the BD1 of Brd2, Brd3, and Brd4 [85]. In numerous AML and prostate cancer cell lines, ABBV-744 showed potent antiproliferative activity, with IC_50_ in the low nanomolar range. Importantly, its greater anticancer activity was also evidenced in prostate tumor xenografts in vivo [84]. At a dose as low as 4.7 mg/kg, ABBV-744 was shown to remarkably suppress tumor growth in vivo with minimal toxicity. Similarly, another rationally designed BD2-selective BET inhibitor (SJ432) (Figure 5D) was recently shown to produce remarkable antitumor potency at a relatively low dose of 15 mg/kg in various neuroblastoma models without notable toxic effects in vivo [86]. Both ABBV-744 and SJ432 exhibited substantial improvement from JQ1 (the prototype pan-BET inhibitor), where a dose of 50–100 mg/kg was generally needed to produce antitumor effects in vivo. Interestingly, the findings from the two BD2-selective BET inhibitors are in discrepancy with the report from Gilan et al. which suggested that only BD1 is required for the anticancer effects of the BET inhibitor [26]. It is likely that BD1 and BD2 have distinctive roles in propelling cancer development in different cancer types. 

The toxic effect of BD1- and BD2-selective BET inhibitors was further investigated in a most recent study on radiation-induced profibrotic fibroblast response [83]. Radiotherapy is known to induce fibrosis as one of its various adverse effects in cancer patients. To this end, radiation was reported to induce the aberrant regulation of profibrotic genes via epigenetic mechanisms. The BD2-selective agent (GSK046) was found to effectively inhibit radiation-induced profibrotic marker genes without showing significant cytotoxic effect [83]. Thus, BD2-selective targeting agents may be used to prevent radiotherapy-induced fibrosis. 

### 3.3. Bivalent BET Inhibitors 

Bivalent BET inhibitors have been designed to bind to two bromodomains simultaneously to slow down dissociation kinetics and to enhance cellular potency [87]. This was supported by the initial findings from the first few reported bivalent BET inhibitors (MT1 (Figure 6A) and biBET (Figure 6B)) that they are remarkably more potent than their monovalent counterparts in terms of binding affinity and the cellular inhibition of BRD4 [87,88]. AZD5153 (Figure 6C) is a recently reported bivalent BET inhibitor which interacts with both the BD1 and BD2 of Brd4 [89]. AZD5153 was shown to displace Brd4 from the chromatin at a lower concentration than that achieved by I-BET762 (a monovalent JQ1 analog). In various hematological cancer models in vitro, AZD5153 exhibited a potent cancer growth suppressive effect with a GI_50_ < 25 nM. Promising antitumor effects were also reported for AZD5153 against a few hematological and thyroid tumor models in vivo at doses as low as 5–10 mg/kg [89,90].

While bivalent BET inhibitors exhibit better activity than their monovalent counterparts, most of them showed unsatisfactory pharmacokinetic properties (limited metabolic stability by liver enzymes) due to the presence of long chain linkers. Recently, a series of novel bivalent BET inhibitors with short and hydrophilic linkers have been developed [91]. They exhibited good anticancer activity both in vitro and in vivo, which could be administered orally. 

The development of bivalent BET inhibitors may also allow the selective targeting of individual BET proteins. Indeed, most previously reported BET inhibitors lack intra-BET selectivity. They cannot differentiate between different BET member proteins (Brd2, Brd3, Brd4, and Brdt) due to the high sequence homology and structural similarity of the BET bromodomains [92]. The BD1/BD2-selective BET inhibitors described in the previous section are capable of specifically targeting the two individual bromodomains (BD1 or BD2), but they still maintain the pan-BET properties [26,84,93,94]. 

Brdt is only expressed in the testes, and it is a validated male contraceptive drug target [95,96]. Brdt has attracted attention for the design of novel non-hormonal male contraceptives [93,97]. On the other hand, Brd4 is ubiquitously expressed, and it is a promising anticancer target. Previous BET inhibitors cannot differentiate between Brdt and Brd4. To this end, although Brd4 is a promising anticancer target, its inhibition is undesirable in male contraceptive drugs. Therefore, new strategies are needed to increase the selectivity for Brdt. Guan et al. hypothesized that the selectivity for Brdt inhibition could be achieved with bivalent inhibitors capable of inducing and stabilizing different conformational states of Brdt and Brd4 [98]. The team modified the nonselective isoquinolinone and diaminopyrimidine monovalent BET inhibitors and designed symmetrical and asymmetrical bivalent inhibitors that exhibited greater affinity and selectivity for Brdt [98]. 

### 3.4. Tyrosine Kinase and BET Dual Inhibitors

In cancer therapy, drug combinations are usually used to increase the therapeutic effect, reduce toxicity, and delay the emergence of drug resistance. The synergistic anticancer effect of BET and tyrosine kinase inhibitors has been reported for a number of cancer types including AML, breast cancer, lymphoma, and osteosarcoma [99,100,101,102,103]. Recently, a few tyrosine kinase inhibitors used for molecular-targeted chemotherapy have been shown to inhibit the acetylated lysine binding site(s) of Brd4 [104,105,106]. The design of single-molecule dual inhibitors may be particularly beneficial because drug combinations may give rise to additive toxicity and subsequently require the dose reduction of the individual drugs that could compromise the therapeutic effect [107]. Dual Brd4-tyrosine kinase inhibitors were designed to specifically target cancers that depend on Brd4 functionality and aberrant oncogenic kinase activity [106].

The dianilinopyrimidine molecular scaffold has been widely used for the design of novel tyrosine kinase inhibitors. Ember et al. reported that this scaffold is also capable of selectively and effectively inhibiting BET proteins [26]. The first-in-class dual Brd4-kinase inhibitors were designed by using the dianilinopyrimidine scaffold of the JAK2/FLT3 inhibitor TG101348 (Figure 7A) [108]. The novel dual Brd4-kinase inhibitors were shown to selectively and potently inhibit Brd4 bromodomains and also a set of tyrosine kinases (JAK2, FLT3, RET, and ROS1). In a panel of cancer cell lines and patient samples of JAK2-driven myeloproliferative neoplasm, the dual inhibitors exhibited superior anticancer activities to JQ1 and TG101348. A few other dual BET-kinase inhibitors, including PI3K-BET inhibitors, CDK-BET inhibitors, and MAPK-BET inhibitors [109,110,111], have also been recently reported. An excellent review on the design and application of dual Brd4-kinase inhibitors has been recently published [112].

However, drug resistance is an unresolved problem for cancer treatment using molecular kinase inhibitors despite the encouraging initial response. Common drug resistance mechanisms include the acquisition of additional oncogenic site mutations and the upregulation and activation of compensatory kinases and other signaling molecules. The circumvention of drug resistance requires attacking cancer cells at multiple levels, usually by drug combinations of individual drugs targeting specific signaling proteins. Dual Brd4-kinase inhibitors may be particularly useful for treating cancers with acquired resistance to single-activity tyrosine kinase or BET inhibitors. 

Inspired by the encouraging efficacy of the dual Brd4-kinase inhibitors, Burgoyne et al. recently reported a rationally designed triple-activity inhibitor that concomitantly inhibits three critical targets (namely, CDK4/6, PI3K, and BRD4) according to known synthetic lethality relationships in cancer [113]. The first triple-action single-molecule inhibitor SRX3177 (Figure 7B) exhibited excellent oncogenic kinases and BET bromodomain selectivity. SRX3177 also showed a marked cytotoxicity toward multiple cancer types in vitro, but it was not toxic to normal cells. Further investigation in resistant cancer models is warranted to illustrate the therapeutic potential of the triple-acting inhibitors. 

### 3.5. Brd4-Selective Inhibitors 

As mentioned above, the BET family proteins consist of four members: Brd2, Brd3, Brd4, and Brdt. Brdt is only expressed in the testes, whereas the other three members are ubiquitously expressed in different tissues in the body. Brd2 and Brd3 are highly similar, and they bind to acetylated histones and promote gene transcription. Besides binding to histone, Brd3 also interacts with several transcriptional factors and regulates immunological response. It is noteworthy that Brd2 may possess both activation and inhibitory functions because it can bind with both transcription activator and inhibitor proteins [114]. Therefore, a Brd4-selective inhibitor may be more preferable to the pan-BET inhibitors for cancer therapy [115,116]. A few representative Brd4-selective inhibitors are described below.

#### 3.5.1. AZD5153 

AZD5153 (Figure 8A) is a novel Brd4-selective inhibitor with a distinct binding mode from other previously described Brd4 inhibitors [117]. It binds simultaneously to two Brd4 bromodomains, thus making it extremely potent in displacing Brd4 from chromatin [93]. This unique biophysical property allows AZD5153 to potently inhibit proliferation and induce apoptosis of numerous cancers, including colorectal cancer in preclinical models [118]. Moreover, AZD5153 was found to significantly inhibit the expression of Wee1 and impair the G2/M cell cycle checkpoint. It was further shown to enhance the anticancer effect of the DNA-damage-related PARP inhibitor (BMN673) both in vitro and in vivo, and with limited toxicity [118]. 

#### 3.5.2. dBET57

dBET57 (Figure 8B) is a potent small-molecule PROTAC-based Brd4-specific degrader, which recruits the proteasome system to specifically target the BD1 bromodomain of Brd4 protein for degradation via ligand-dependent CRBN/Brd4 interactions [119]. In neuroblastoma, the most common solid tumor of the neural crest cell origin in children, dBET57 was recently shown to target Brd4 ubiquitination, inhibit the super-enhancer-related genes *ZMYND8* and *TBX3*, and exhibit a potent anticancer effect both in vitro and in vivo [120]. 

#### 3.5.3. MZ1 

Zengerle et al. employed the small-molecule PROTAC approach to design a series of novel compounds to selectively induce Brd4 degradation [121]. The PROTACs were designed that tether an ester-hydrolyzed JQ1 analog to a ligand for the E3 ubiquitin ligase von Hippel–Lindau protein (VHL). One of the small-molecule PROTAC compounds (termed compound MZ1 (Figure 8C)) was shown to trigger a rapid, durable, and selective degradation of Brd4 over Brd2 and Brd3. In the breast cancer cell line HeLa, a preferential alteration of Brd4-dependent genes (including *MYC*, *p21*, and *AREG*) but not other Brd4-independent genes (such as *FAS*, *FGFR1*, and *TYRO3*) was induced by MZ1 relative to JQ1, which was consistent with the selective suppression of Brd4 by MZ1 [121]. The Brd4 selectivity may be generated by the preferential direct interaction or reduced steric hindrance between VHL and Brd4 compared to Brd2/Brd3, thus promoting the formation of a VHL–PROTAC–Brd4 ternary complex for protein degradation [121]. 

#### 3.5.4. NHWD-870

Among the four BET proteins, Brd4 is known to be highly enriched in super-enhancers that promote the expression of factors including c-MYC to support cancer growth and development. NHWD-870 (Figure 8D) was developed recently as a potent and selective BET inhibitor targeting Brd4 [122]. It was shown to be more potent than the three other major BET inhibitors currently in clinical development (including BMS-986158, OTX-015, and GSK-525762). NHWD-870 adopts a distinct core structure from the early-generation BET inhibitors (JQ1, OTX-015, I-BET762, and CPI-0610), and it binds well with the Brd4 bromodomains. Besides directly suppressing tumor growth via the inhibition of Brd4 and c-MYC transcription, NHWD-870 also suppressed the proliferation of tumor-associated macrophages (TAMs) through reducing the expression and secretion of the macrophage colony stimulating factor CSF1 by tumor cells [122]. By inhibiting the tumor-infiltrating TAMs, NHWD-870 exhibited potent anticancer effect in hematological cancers and also solid cancers in xenograft and syngeneic animal models [122]. Mechanistically, the inhibition of CSF1 by NHWD-870 was associated with the suppression of Brd4 and its target HIF-1α. As TAMs are known to suppress antitumor immunity by preventing the activation of dendritic cells, cytotoxic T lymphocytes, and natural killer cells, the inhibition of TAMs has been exploited as a novel means to promote the efficacy of anticancer therapy [123].

#### 3.5.5. ZL0513 

Angiogenesis is a key pathological process driving cancer development. New capillaries are formed in order to facilitate the delivery of oxygen, nutrients, and growth factors to the growing tumor mass. Brd4 is known to promote RNA polymerase II (RNAPII) pause release and drive angiogenesis [124]. While the pan-BET inhibitor (+)-JQ1 was reported to inhibit angiogenesis, the role of Brd4 and that of its bromodomains (BD1 and BD2) in angiogenesis remains elusive because of the nonselective inhibition of (+)-JQ1 on all BET family members. Recently, a potent and selective Brd4 inhibitor (ZL0513) (Figure 8E) was identified by screening an in-house compound library targeting BET family proteins [125]. ZL0513 was shown to produce significant anti-angiogenic effects in chick embryo chorioallantoic membrane and yolk sac membrane models [126]. Mechanistically, ZL0513 was found to inhibit the phosphorylation of the AP-1 transcription factors c-jun and c-fos to suppress angiogenesis. This novel Brd4 inhibitor represents an important pharmacological tool for elucidating the roles and biological functions of Brd4 and its BD bromodomains in angiogenesis. Moreover, it may also serve as a lead compound that could target the vasculature in angiogenesis-dysregulated human diseases including cancer [126]. 

## 4. Toxicities of BET Inhibitors

The therapeutic efficacy of BET inhibitors is often hindered by dose-limiting toxicities as they target any proteins containing bromodomains. Most recently, a systematic review about the safety, efficacy, and pharmacodynamics of 12 BET inhibitors in clinical trials has been conducted [127]. All BET inhibitors exhibited similar toxicity profiles, with gastrointestinal toxicity (diarrhea, nausea, and vomiting), thromobocytopenia, anemia, fatigue, and hyperbilirubinemia most commonly observed in clinical studies [128]. The undesirable toxicity is considered off-tissue but on-target inhibition or degradation of the bromodomains. 

In fact, thrombocytopenia is the most common dose-limiting toxicity observed in cancer patients on monotherapy with BET inhibitors [127]. Importantly, over half of the affected patients experience severe thrombocytopenia, which necessitates dose reduction or therapy termination [127]. In a *BRD4*-knockout mouse model, it has been shown that Brd4 is required for hematopoietic stem cell expansion and progenitor cell development [129]. Based on the KEGG enrichment analysis from the published clinical data, apoptosis induced by the BET inhibitors may be the major mechanism contributing to thrombocytopenia [127]. Dose modification and/or platelet transfusion are usually considered in patients on BET inhibitor treatment to prevent fatal complications such as cerebral and gastrointestinal hemorrhage. 

Regarding the non-hematological adverse effects, nausea, diarrhea, fatigue, dysgeusia, and decreased appetite are the major ones [127]. The mechanism for the significant gastrointestinal toxicity has been investigated in a transgenic RNAi mouse model [130]. Sustained BET inhibition due to genetic silencing was shown to cause reversible hyperplasia, alopecia, and stem cell depletion in the small intestine of *BRD4*-depleted mice [130]. Moreover, there was remarkable impairment in the intestine regeneration of the BRD4-inhibited mice after irradiation [130]. Therefore, extra precaution is needed when BET inhibitors are combined with radiation therapy in the treatment of cancer. 

## 5. Combination of BET Inhibitor with Other Chemotherapeutic Modalities for Cancer Treatment 

As BET inhibitors exhibit only limited anticancer efficacy as a single agent alone, the combination of BET inhibitors with other chemotherapeutic drugs has been investigated as a means to enhance the therapeutic outcome [131].

### 5.1. Combination with Conventional Chemotherapeutic Drugs 

The combination of BET inhibitors and paclitaxel or cisplatin has been investigated in the treatment of non-small-cell lung cancer (NSCLC) [132]. The drug combination was found to suppress BET expression. The synergistic anticancer effect was associated with the inhibition of autophagy and potentiation of apoptosis [132]. Fueled by the synergistic combination effect of paclitaxel and JQ1, a novel nanoparticle-based formulation has been developed to encapsulate both drugs together in Zein nanoparticles [133]. This novel nanoformulation was reported to exhibit a superior anticancer effect to either drug alone in triple-negative breast cancer cells in vitro [133].

### 5.2. Combination with Epigenetic Drugs 

The anticancer effect of BET inhibitors has been attributed to the transcriptional suppression of the *MYC* oncogene. In a MYC-transgenic lymphoma model, Bhadury et al. reported that BET inhibition indeed exhibited broader transcriptional effects by modulating numerous transcription factor networks [134]. Genes induced by BET inhibitors were surprisingly similar to those induced by histone deacetylase (HDAC) inhibitors [134]. Therefore, the anticancer effect of BET inhibitors in Myc-overexpressing lymphoma may be derived partly by the induction of HDAC-silenced genes. Targeting the genetic link between BET inhibitors and HDAC inhibitors was proposed to give rise to a synergistic anticancer effect. In fact, BET inhibitors have been shown to synergize with HDAC inhibitors in several cancer types (including lymphoma, testicular germ cell tumor, melanoma, and pancreatic ductal adenocarcinoma) and produce a canonical apoptotic response [134,135,136,137]. The downregulation of oncogenic and anti-apoptotic factors, including c-MYC and BCL-2, was believed to contribute to the synergistic anticancer effect from the combination of HDAC and BET inhibitors [138,139]. On the other hand, a synergistic anticancer effect was also reported in the combination of BET inhibitors and various other epigenetic drugs (including JQ1-panobinostat (HDAC inhibitor) [140], OTX015-azacitidine (demethylating agent) [141], and I-BET151-SGC0946 (histone H3K79 methyltransferase inhibitor) [142]) in leukemia cell lines and mouse models. 

### 5.3. Combination with Molecular-Targeted Agents

Bruton’s tyrosine kinase (BTK) is a cytoplasmic tyrosine kinase that is often dysregulated in B-cell malignancies [143]. The tyrosine kinase plays a critical role in regulating B-cell proliferation and chemotaxis. BTK inhibitors represent a promising novel class of molecular-targeted agents for B-cell malignancies including non-Hodgkin lymphomas and chronic lymphocytic leukemia. To this end, the combination of BTK inhibitors with BET inhibitors was found to be synergistic in neoplastic B cells [51]. The simultaneous inhibition of BET and BTK in malignant B cells was shown to downregulate MYD88 expression, subsequently suppressing its downstream signaling through NF-kB, Toll-like receptors and the JAK/STAT pathway [51]. 

In PI3K- and MYC-driven metastatic breast cancer, resistance to the class I PI3K inhibitor was caused by the feedback activation of tyrosine kinase receptors AKT, mTOR, and MYC. While BET inhibitors alone did not exhibit significant anticancer activity in this tumor type, the combination of BET and PI3K inhibitors was shown to produce a synergistic anticancer effect both in vitro and in vivo [101]. The apparent circumvention of resistance to PI3K inhibitors advocates their translational application in clinical trials. 

FLT3 is a receptor tyrosine kinase expressed by immature hematopoietic progenitor cells. FLT3 mutations are the most common genetic abnormality driving AML development. A few small-molecule FLT3-targeted inhibitors have been approved for AML treatment. Despite an initial clinical response to FLT3 inhibitors, drug resistance through secondary FLT3 mutations and the activation of parallel signaling pathways is posing a significant hindrance to AML disease management [144]. Recently, the combination of JQ1 and quizartinib (FLT3 inhibitor targeting the internal tandem duplication mutation) has been shown to apparently overcome FLT3 inhibitor resistance by attenuating c-MYC, BCL2, and CDK4/6 expression [100]. 

### 5.4. Combination with Cyclin-Dependent Kinase (CDK) Inhibitors 

CDKs play a critical role in regulating cell cycle progression, and they are dependent on cyclins for their activity. CDKs and various cyclins are often overexpressed in cancer, thus making them attractive molecular targets for cancer treatment. On the other hand, CDKs (in particular CDK1 and CDK2) are known to promote MYC protein stability by inducing its phosphorylation at Ser-62 [145]. In MYC-driven medulloblastoma, the combined BET bromodomain and CDK2 inhibition was shown to inhibit MYC expression differentially by targeting MYC transcription (by the BET inhibitor JQ1) and MYC protein stabilization (by the CDK2 inhibitor Milciclib), respectively [146]. Thus, the combination of BET and CDK2 inhibitors was shown to produce pronounced a synergistic anticancer effect, which was accompanied by significant cell cycle arrest and massive apoptosis [146]. 

### 5.5. Combination with Proteasome Inhibitors

Proteasome inhibitors (bortezomib, carfilzomib, and ixabomib) are currently the mainstay of the treatment for hematological cancers, particularly multiple myeloma. However, they have limited efficacy against solid tumors, and the emergence of drug resistance to proteasome inhibitors is almost inevitable. To this end, BET inhibitors (JQ1, I-BET762, and I-BET151) have been recently reported to produce a synergistic anticancer effect when combined with carfilzomib in multiple solid tumor cancer lines [147]. BET inhibitors were shown to attenuate the Nrf1-mediated induction of proteasome genes in response to proteasome inhibition, thus inhibiting the rebound of proteasome activity in cancer cells following proteasome inhibitor treatment [147]. On the other hand, bortezomib resistance in mantle cell lymphoma is associated with the overexpression of IRF4, BLIMP-1, and MYC, and the plasmacytic differentiation phenotype [148]. Thus, the combination of lenalidomide (a novel proteasome inhibitor) and CPI-203 (JQ1 analog with superior bioavailability) was found to be highly synergistic in bortezomib-resistant myeloma cells [149]. 

### 5.6. Combination with Immunotherapeutic Drugs 

Programmed cell death receptor-1 (PD-1) is an inhibitory receptor expressed on activated T cells, B cells, and natural killer cells, which normally function to blunt the immune response. PD-1 is engaged by its major ligand PD-L1, which is expressed in tumor cells and infiltrating immune cells, to suppress the T-cell-mediated cancer-killing effect. Cancer cells can evade T-cell immune responses by driving T-cell exhaustion, which is characterized by the overexpression of multiple inhibitory receptors or immune checkpoint molecules including PD-1. The blockade of the PD-1 pathway is an effective immunotherapeutic strategy for cancer treatment. However, the response rate to the PD-1 inhibitor is limited, and many patients cannot achieve a durable clinical response. There has been extensive research for novel strategies that can enhance anticancer immunity. Recently, the BET inhibitor JQ1 was reported to rescue PD-1-mediated T-cell exhaustion in AML [150]. JQ1 was shown to suppress the expression of PD-1 and promote the secretion of cytokines in T cells from AML patients [150]. Mechanistically, BRD4 binds to the *PDCD1* (encoding PD-1) promoter. JQ-1-treated T cells exhibited downregulation of PD-1 expression [150]. 

## 6. Resistance to BET Inhibitors 

The mechanisms leading to BET inhibitor resistance are multifactorial. Interestingly, none of the reported resistance mechanisms are related to genetic aberrations of the bromodomains (i.e., BRD2/3/4 mutations). In ovarian cancer, prolonged treatment with BET inhibitors has been reported to cause receptor tyrosine kinase reprogramming and subsequently resistance to BET inhibitors [151]. In colorectal cancer, activated interleukin 6/8-Janus kinase 2 signaling was known to promote Brd4 phosphorylation. To this end, phosphorylated Brd4 becomes more stable and binds to BET inhibitors with lower affinity, thus contributing to BET inhibitor resistance [152]. In AML and pancreatic cancer cells, the compensatory upregulation of MYC via the WNT pathway was reported to reduce the responsiveness of the cancer cells to BET inhibitors [153,154]. Moreover, JQ1-resistant AML cells did not undergo apoptosis but switched to pro-survival autophagy [155]. JQ-1-induced autophagy in the resistant AML cells was associated with the upregulation of Beclin 1, increased LC3-II expression, and accumulation of autophagosomes, which was independent of mTOR signaling [155]. In triple-negative breast cancer, resistance to BET inhibitors was mediated by a bromodomain-independent mechanism [156]. In prostate cancer, the loss-of-function mutation of SPOP (an E3 ubiquitin ligase of Brd4) has been shown to confer resistance to BET inhibitors by impairing ubiquitination-mediated Brd4 degradation [157,158]. The maintenance of MYC expression was also reported to promote de novo resistance to BET inhibitors in castration-resistant prostate cancer [159]. Following prolonged treatment with CRBN- or VHL-based BET-targeting PROTACs, acquired resistance has been shown to emerge due to genomic alternations that compromise core components of the CRBN or VHL E3 ligase complexes [79]. On the other hand, the hyperphosphorylation of Brd4 due to the downregulation of the phosphatase PP2A [156] and elevated expression ratio of BCL2L1/BCL-XL [156] has also been reported to contribute to BET inhibitor resistance in triple-negative breast cancer. In pancreatic cancer, prolonged treatment with JQ-1 has been shown to trigger rebound increase in the BET inhibitor target genes including *FOSL1* and *HMGA2* [154]. 

## 7. Potential Biomarkers Predicting the Activity and Resistance of BET Inhibitors

### 7.1. Predictive Biomarkers for the Anticancer Effect of BET Inhibitors 

#### 7.1.1. Presence of BRD4-–NUT Fusion 

NUT carcinoma (NC) is a rare but very aggressive subtype of squamous carcinoma harboring the characteristic genetic rearrangement involving the *NUTM1* gene. The most common *NUTM1* fusion partner is *BRD4*, constituting up to 75% of all NC cases [160]. The *BRD4–NUTM1* fusion gene is driven by the *BRD4* promoter. The BRD4–NUTM1 fusion protein was known to tether *NUTM1* to acetylated chromatin via the Brd4 bromodomains. Subsequently, *NUTM1* recruited the histone acetyltransferase EP300 to increase the acetylation of the surrounding chromatin to create a highly acetylated mega-domain, where proliferation genes (such as *MYC* and *TP63*) are induced to promote cancer proliferation [48]. Owing to the hallmark *NUTM1* rearrangement of NC, diagnosis is usually made by immunohistochemical staining that recognizes the NUTM1 protein in the tumor tissue. Additional confirmation can be made by identifying the fusion partner of *NUTM1* with fluorescence in situ hybridization (FISH) using split-apart probes or next-generation sequencing [161]. It is noteworthy that the *BRD4–NUTM1* fusion (when compared with other fusion partners such as *NSD3* or *BRD3*) is associated with worse overall survival [162]. 

#### 7.1.2. Expression of the BET Family of Bromodomain Proteins

The general consensus is that Brd4 is the primary molecular target of most BET inhibitors. Therefore, tumor types that are dependent on Brd4 for survival will likely be the ones more sensitive to Brd4 inhibition. In fact, it has not been fully established whether the overexpression of the BET-family proteins affects the drug sensitivity to BET inhibitors. To this end, most BET inhibitors developed to date target all members of the BET family (i.e., Brd2, Brd3, Brd4, and Brdt) [49]. The various BET proteins may exhibit distinct and overlapping oncogenic driver activity in different cancer types. More research is needed to elucidate the cancer cell-type-specific oncogenic properties of the different BET proteins and to evaluate the effect of their relative expression profile of sensitivity to BET inhibitors. 

#### 7.1.3. MYC Amplification 

MYC is a master transcriptional regulator controlling numerous cellular processes including cell proliferation, differentiation, and survival [163]. In preclinical studies, *MYC* is a well-known Brd4 target gene in multiple cancer types. BET inhibitors inhibit *MYC* transcription and trigger genome-wide suppression of *MYC*-dependent target genes to elicit their anticancer effect [164]. It was therefore widely believed that an elevated *MYC* expression or *MYC* amplification would enhance the sensitivity to BET inhibitors [46,84,86]. However, in clinical trials investigating the BET inhibitor OTX015 in multiple myeloma, AML, and diffuse large B-cell lymphoma, *MYC* amplification was found incapable of predicting the drug response to the BET inhibitor [165,166]. There have been arguments that high concentrations of BET inhibitors used in preclinical studies could have off-target effects [130]. They may be exacerbated in MYC-overexpressing cells. Alternatively, high concentrations of BET inhibitors may suppress MYC by BET-independent pathways. To ascertain the validity of *MYC* amplification as a predictive biomarker for BET inhibitors, further clinical studies will be needed to demonstrate a significant clinical response in patient cohorts with *MYC*-amplified tumors, which is accompanied by downregulation of *MYC* after the drug treatment. On the other hand, while *MYC* amplification was considered a major mechanism by which Myc is dysregulated in cancer, the post-translational modification of Myc (particularly phosphorylation) also plays an important role in regulating Myc protein stability and activity [167]. The status of Myc phosphorylation in tumoral tissues may also be used as a potential biomarker to predict sensitivity to BET inhibitors. 

### 7.2. Biomarkers of Resistance 

#### 7.2.1. Aberrant Activation of Receptor Tyrosine Kinase (RTK) Signaling Pathways 

Mutations of RTKs and the aberrant activation of their intracellular signaling cascade are associated with cancer development and drug resistance to various anticancer drugs. In an extensive screening study utilizing genome-scale, pooled lentiviral open reading frame (ORF), and CRISPR knockout rescue techniques, the upregulation of RTKs, activation of PI3K signaling pathway, and a global enhancer remodeling were generally observed in various in vitro neuroblastoma cell models exhibiting acquired resistance to BET inhibitors [168]. A remarkable enrichment of the H3K27Ac histone modification was observed at the enhancer elements of the resistance genes that are transcriptionally upregulated [168]. Importantly, a significant increase of Brd4 binding on the active enhancers was observed. Moreover, PI3K inhibitors were identified to produce the most pronounced synergistic anticancer effect with BET inhibitors in the resistant neuroblastoma cells [168]. In another study investigating ovarian cancer cells with acquired resistance to BET inhibitors, the reprogramming of the kinome with a remarkable activation of the PI3K and RAS signaling pathways was observed [151]. Recently, *KRAS* mutations were also identified as novel resistance biomarkers for BET inhibitors by the genetic analysis of a large set of ~230 cancer cell lines, spanning both hematological and solid cancers, after treatment with a new BET inhibitor GSK525762 (molibresib) in a Phase 1 clinical trial [59]. Therefore, the evaluation of PI3K and RAS signaling molecules as biomarkers of resistance may provide the guidance for the rational combination of BET inhibitors and targeted therapeutic drugs to overcome drug resistance to BET inhibitors. 

#### 7.2.2. Aberrant Ubiquitination and Degradation of BET Proteins 

The stability of BET proteins has been reported to affect the drug sensitivity to BET inhibitors. The speckle-type POZ protein (SPOP) is known to regulate the ubiquitination and degradation of BET proteins [169]. In prostate cancer, mutations within the substrate binding site of SPOP were reported to mediate intrinsic resistance to BET inhibitors in vitro and in vivo [157,158]. The BET protein degradation was substantially reduced in SPOP-mutated prostate cancer cells, leading to higher cellular BET protein expression [170]. The retardation of BET protein degradation was confirmed to be a mechanism leading to BET inhibitor resistance because the depletion of BET proteins could re-sensitize the resistant cancer cells to the BET inhibitor [157,158]. Interestingly, other SPOP mutations (such as R121Q) were reported to promote the association of SPOP and BET proteins, thus increasing BET ubiquitination but reducing BET protein stability to enhance the sensitivity to BET inhibitors [170]. Therefore, the specific SPOP mutations and their functional effect on the BET proteins could allow them to serve as biomarkers of both resistance and sensitivity to BET inhibitors. 

## 8. Targeting at the Extra-Terminal (ET) Domain of Brd4 

### 8.1. Significance of Targeting the BET/Brd4 Extra-Terminal Domain

Although the dual N-terminal bromodomains (BD1 and BD2) are well-characterized within the BET family of proteins, they are also present in more than 10 subfamilies of bromodomain-containing proteins [13,171]. Thus far, bromodomain-targeted agents have shown dose-limiting toxicities (DLT) in clinical trials [165,172,173] because of their effects on other bromodomain-containing proteins, both cancer-specific and normal. Focusing on the bromodomain inhibition of BET proteins may lead to unwanted side effects due to the interference of other proteins containing similar domains. However, BET proteins are distinguished from other bromodomain-containing proteins by a unique C-terminal extra-terminal (ET) domain [17], through which BET proteins are associated with a variety of viral and cellular proteins including Jumonji C-domain-containing protein 6 (JMJD6) [22,23], nuclear receptor-binding SET domain protein 3 (NSD3) [18,19,20,21], viral integrase (IN) from murine leukemia virus (MLV) [174], and others. One of the prominent members of BET proteins, Brd4, has been implicated in multiple cancers including neuroblastoma, oral, breast, lung, and colon malignancies through ET-mediated protein–protein interactions [38,175]. For example, interactions between JMJD6 and the Brd4 ET domain are known to drive the progression of neuroblastoma, oral, breast, lung, prostate, and colon cancers [23,170,176,177,178,179,180,181,182,183]. Interactions of NSD3 with the Brd4 ET domain are required in aggressive midline carcinoma and are essential for AML maintenance [13,19,184]. Furthermore, targeting Brd4 could serve as an avenue to prevent cellular proliferation and T-cell exhaustion in AML patients [150,185]. The role of MLV-based vectors in human cancers was first demonstrated in hematopoietic stem cell (HSC) gene-therapy studies [186,187,188,189] which showed that insertion near the *LMO-2* and *CCND2* proto-oncogenes resulted in leukemic development. This was later linked to the interaction of MLV IN with the Brd4 ET domain, which directed integration toward these proto-oncogenes [190,191,192]. However, there are currently no therapies that directly target the ET domain of BET proteins.

### 8.2. Interaction between Brd4-ET and Other Protein Partners

#### 8.2.1. Interaction between the ET Domain and JMJD6

Jumonji C-domain-containing protein 6 (JMJD6), a Fe(II)- and 2-oxoglutarate (2OG)-dependent oxygenase, interacts with Brd4 protein through the ET domain. Back in 2013, Liu et al. demonstrated that JMJD6 and Brd4 are crucial in the regulation of the promoter-proximal pause release of a significant proportion of transcription units [22]. JMJD6/Brd4 interactions in the context of the active P-TEFb complex also regulate the RNA polymerase II promoter-proximal pause release of a substantial cohort of genes. The structural basis of JMJD6 recognition by Brd4 was solved by Konuma et al. using solution NMR [23]. Brd4 was shown to interact with a JMJD6 sequence (K84-N96) through its ET domain [23]. The JMJD6 peptide adapts an α-helix secondary structure at the conserve hydrophobic core of Brd4-ET and is reinforced by the electrostatic interactions of K91 and R95 in JMJD6 with residues E657, E653, E651, and D650 in the inter-helical α2-α3 loop of the ET domain (annotations of secondary structures are shown in Figure 9). The binding K_d_ between the Brd4-ET and JMJD6 peptide characterized by an isothermal titration calorimetric (ITC) experiment was 158 ± 14 μM, and a double-transfected co-immunoprecipitation experiment also confirmed such an interaction in the cellular context. As shown in Figure 10A, three hydrophobic residues of JMJD6 (W85, L90, and Y94) are buried in the hydrophobic core of Brd4-ET, interacting with I652, I654, and F656 at the ET α2-α3 loop, and also with I622, L630, V633, V634, and I637 from α1 and α2 helices. A large number of intermolecular NOEs were observed on the aromatic ring of W85 of JMJD6 with S619, I622, V633, I637, I654, and F656 of ET; L90 of JMJD6 with I622, Q623, L630, Val633, Ile652, and Ile654 of ET; and Y94 of JMJD6 with I622, L630, V634, and I652 of ET, respectively. No hydrogen bonds between the backbone of JMJD6 and ET were observed, indicating a relatively stable and ordered α-helix was formed when JMJD6 is bound and conformationally adapted by the ET protein. The significance of the interacting residues was also confirmed by mutagenesis studies, which showed that the K_d_ of the W85A_JMJD6_ peptide mutant is 675 ± 66 μM binding with ET, which is four-fold weaker than that of wild-type JMJD6 peptide. The L90A_JMJD6_, K91A_JMJD6_, and R95A_JMJD6_ mutants nearly eliminate the observable binding affinity to the ET domain. The JMJD6 peptide can serve as a good template for further small-molecule or peptidomimetic designs that target the Brd4-ET domain. 

#### 8.2.2. Interaction between the ET Domain and NSD3

Nuclear receptor-binding SET domain protein 3 (NSD3), a *WHSC1L1* gene-encoded histone H3 lysine 36 (H3K36) methyltransferase, belongs to the nuclear receptor Su(var)3–9, enhancer-of-zeste, and trithorax (SET) domain-containing (NSD) family. Shen et al. demonstrated that acute myeloid leukemia (AML) maintenance mediated by Brd4 requires its interaction with NSD3 as an effector [19]. Interestingly, a short isoform of NSD3 lacking catalytic function is also essential in AML progression and directly binds to the Brd4-ET domain, which links Brd4 to the chromatin-remodeling enzyme CHD8 to form a Brd4–NSD3–CHD8 complex. Zhang and co-workers [20] have solved two solution NMR structures (Figure 10B,C) of the ET domain bound with two peptide sequences derived from NSD3 protein, denoted as NSD3_1 (^152^EIKLKITKTIQN^163^) and NSD3_3 (^593^VVPKKKIKKEQVE^605^). Binding K_d_ of NSD3_1 was determined as 140.5 μM by an isothermal titration calorimetric (ITC) experiment. Structure studies showed that the protein–peptide recognition was established by the ^651^EIEID^655^ sequence on β1 strand of the ET domain and ^154^KLKI^157^ sequence on NSD3_1. Electrostatic interactions of K154_NSD3_ and K156_NSD3_ with D655_Brd4_ and E653_Brd4_ served as the major anchors, along with the backbone hydrogen bonds which further stabilize the interaction. Furthermore, L155 _NSD3_ and I157_NSD3_ are well accommodated in the hydrophobic core of the ET domain surrounded by I622, V633, I654, I652, L630, and V634 of Brd4. Alanine scanning on the NSD3_1 peptide further confirmed that K154A_NSD3_ and K156A_NSD3_ mutants completely eliminated the binding, and that L155A_NSD3_ and I157A_NSD3_ mutants also showed significantly decreased binding affinity due to the lack of hydrophobicity. Remarkably, the pattern of alternating positively charged and hydrophobic residues on the NSD3_1 peptide is critical for maintaining binding affinity. The NSD3_3 peptide showed a decreased K_d_ (~500 μM estimated by NMR) due to lacking as conserved of a pattern compared with NSD3_1. Confirmed by the NMR structure, ^596^KKK^598^_NSD3_3_ interacts at a similar region as ^154^KLK^156^_NSD3_1_, but K597_NSD3_3_ is not a hydrophobic residue and leads to an unfavorable clash at the hydrophobic core of the ET domain. Although the secondary structure of the NSD3_1 peptide is different from JMJD6 when bound to the ET domain, the recognition patterns are consistent. Five years later, Aiyer and co-workers [193] further observed more details about NSD3/Brd3-ET interactions. Even though the ET domain in this work is from Brd3 instead of Brd4, the conservation between these two BET family members makes it applicable to investigate the protein interaction features of both for comparison. Solution NMR structures indicated that an expanded sub-region of NSD3 (sequence 100–263) caused dramatically different chemical shift profiles (CSPs) from that of the previously reported NSD3_1 peptide, suggesting the existence of additional uncharacterized interactions in the previous structures. After redefining the boundary of the ET-interacting region, Aiyer et al. [193] reported an updated model (Figure 10D) where the sequence of 154–172 of NSD3 folded back as a beta-sheet at the ET domain, and ^169^ESSL^172^ is positioned next to α1 helix of the ET domain and forms backbone hydrogen bonds with ^154^KLKITK^159^. These models provide extremely helpful information for developing small molecules targeting at the ET domain to disrupt oncogenic protein–peptide recognitions by either pharmacophore-based or peptidomimetic approaches. 

#### 8.2.3. Interaction between the ET Domain and Chromatin-Remodeling Complexes

In addition to JMJD6 and NSD3, the ET domain is also the protein–protein interaction module engaging with a range of chromatin-remodeling complexes. This interaction serves as a potential route of therapy by disrupting specific chromatin-associated protein complexes [84]. Existing research has demonstrated that the ET domain can interact with the nucleosome-remodeling and deacetylation (NuRD) complex, where CHD4 serves as the ATP-driven DNA translocase (as well as with the BAF and PBAF complexes, for which remodeling is driven by SWI2/SNF2-type ATPase, BRG1, or BRM [194,195,196,197]). Wai et al. [198] elucidated the structural basis of the Brd3-ET domain binding with a short peptide from BRG1 and CHD4 (Figure 11A,B), respectively, in which the recognition pattern is very consistent with that of NSD3. The ET binding motif of the CHD4 protein maps to a disordered region and binds at the same location of the ET domain as JMJD6 and NSD3. ^293^PLKIKL^298^ sequence of CHD4 adapts a linear conformation upon engaging with Brd3-ET, and forms electrostatic interactions by K295_CHD4_ and K297_CHD4_ with residues D617_Brd3_, E615_Brd3_, and E613_Brd3_. Other residues including L294_CHD4_, I296_CHD4_, and L298_CHD4_ are buried in the hydrophobic core of Brd3-ET. The same scenario also applies to the interaction between Brd3-ET and BRG1, where the electrostatic interactions were carried out by K1594_BRG1_, K1596_BRG1_, and K1598_BRG1_ with residues D617_Brd3_, E615_Brd3_, and E613_Brd3_, and hydrophobic contacts were maintained by V1595_BRG1_, I1597_BRG1_, and L1599_BRG1_. Therefore, even though the binding partners with the ET domain of Brd4 or Brd3 are from different protein families with various functions, the recognition pattern remains conserved. 

#### 8.2.4. Interaction between the ET Domain and Viral Protein 

In addition to interacting with endogenous proteins, the BET protein ET domain also engages with viral proteins (reviewed in [199]). Retrovirus replication requires the integration of reverse-transcribed viral DNA into the chromosome of the host. For example, the lentiviruses, such as human immunodeficiency virus type 1 (HIV-1), favor integration at active transcriptional units [200]; the gamma retroviruses, such as murine leukemia virus (MLV), favor integration at transcription start sites [191,201]. Larue et al. [192] revealed a bimodal mechanism of Brd4-mediated MLV integration and showed that MLV integrase (IN) interacts with Brd4 with the K_d_ of 58.37 ± 9.57 nM. Crowe et al. [174] reported that the recognition of Brd4 and MLV-IN was mediated by the interaction between the Brd4-ET domain and a 17-mer peptide from MLV IN (Figure 11C), termed the ET-binding motif (EBM). Solution NMR structure has shown that MLV-EBM folded as a beta-sheet at the hydrophobic core of the ET domain. The interaction pattern is quite consistent with NSD3, i.e., a LKIRL sequence with two basic residues can engage with the beta1 sheet of the ET domain mainly through the electrostatic interactions of ^651^EIEID^655^_Brd4_. Likewise, several residues from MLV-EBM, including L399, I401, V392, L403, and W390, are positioned within the hydrophobic core of the ET domain (Figure 11D). In addition to the NMR experiments, affinity pull-down experiments further support that the acidic and hydrophobic residues of the ET domain are critical for the binding and potentially linked to its phosphorylation [174,192,202]. Based on the solved structure model, Xing et al. [203] reported that ^399^LKIRL^403^ is a critical amino acid (or hot-spot) sequence that carries out the MLV-EBM/Brd4-ET interactions. Global docking, molecular dynamics (MD) simulation, and MM/GBSA energy calculations supported that mutating any of the amino acid can cause an unfavorable interaction mode and decrease in binding energy. The pentapeptide LKIRL is sufficient to directly bind the Brd4-ET domain and reduce cellular proliferation of an AML cell line, which could serve as a good template for peptidomimetic drug design and pharmacophore-based screening. Zhang et al. [182] also reported an NMR structure of Brd3-ET bound with a peptide from Kaposi’s sarcoma-associated herpesvirus latency-associated nuclear antigen (LANA) (Figure 11E) with a sequence of ^1133^QSSIVKFKKPLP^1144^, which again follows the similar interaction pattern of the ^1137^VKF^1139^ segment closely engaging with the ET protein backbone and ^651^EIEID^655^ strand. Taken together, these different interacting partners all maintain a similar mode of engagement and bind with the ET domain through electrostatic interactions of the side chain, hydrogen-bond of the backbone, and the hydrophobic interaction within the core groove of the ET domain. 

## 9. Further Perspectives 

BET inhibitors represent an important class of novel anticancer drugs. While the antitumor potential of BET inhibitors has been confirmed in recent clinical trials, their efficacy as single agents seems to be limited. All reported BET inhibitors work by blocking the recognition of the histone acetylation marks by the bromodomains. Therefore, dose-limiting toxicity due to the targeting of other off-target bromodomain proteins is severely hindering the clinical efficacy of BET inhibitors. Moreover, primary and acquired drug resistance has already been reported for BET inhibitors due to multiple mechanisms. To date, no BET inhibitor has received regulatory approval yet for clinical application. A more thorough understanding about BET protein biology, selectivity of BET inhibitors, drug pharmacokinetics, and pharmacodynamic biomarkers is needed to fully unleash the anticancer potential of BET inhibitors. 

While more potent and selective bromodomain inhibitors have been reported, the diverse and unknown functions of bromodomain-containing proteins pose challenges to the systematic evaluation of the cellular effect and selectivity of the inhibitors. Given the structural similarity between different bromodomains, a robust method will be needed to examine the selectivity of new bromodomain inhibitors to avoid off-target effects. Recently, Li et al. have developed a photoaffinity probe (photo-bromosporine (photo-BS)) for the labeling of endogenous bromodomain-containing proteins in living cells [204]. The photo-BS-labeled bromodomains will then be selectively targeted by different bromodomain inhibitors. When coupled with proteomics analysis, the method may be used to investigate the selectivity of new bromodomain inhibitors for their endogenous bromodomain-containing proteins in living cells [204]. 

All reported BET inhibitors work by blocking the recognition of the histone acetylation marks by the bromodomains. Therefore, dose-limiting toxicity due to the targeting of other off-target bromodomain proteins is severely hindering the clinical efficacy of BET inhibitors. To this end, BD1- and BD2-selective BET inhibitors have been developed. They were shown to be better therapeutic strategies for cancer and inflammatory diseases, respectively [26]. Moreover, toxicity associated with single-BD inhibition by BET inhibitors was also found to be considerably milder than that of pan-BD BET inhibition [84]. However, the advantages of BD1/BD2-selective BET inhibitors over pan-BD BET inhibitors remains to be demonstrated in the clinical setting. Apart from the bromodomains, BET proteins also contain a C-terminal ET domain. Importantly, the ET domain is also associated with a variety of cellular proteins such as NSD3 [20] and JMJD6 [22], which are implicated in the development of AML and lung/colon cancers, respectively. To this end, there is currently no investigational therapeutic approach reported to target the ET domain of BET proteins, which may represent a new avenue for the development of novel therapies against BET/Brd4-mediated cancers. On the other hand, among the four BET family of proteins, Brd4 has been the most extensively studied, whereas the relative role of Brd2/Brd3 in cancer cell survival still remain elusive. In recent years, Brd4-selective BET inhibitors have caught a lot of attention. It is desirable to establish the clinical benefit of Brd4-selective BET inhibitors over pan-BET inhibitors in individual cancer types. 

Numerous clinical trials have been conducted to evaluate the efficacy of BET inhibitors in treating both hematological and solid cancers (Table 2). However, except the ones for NUT midline carcinoma (NMC), most other clinical trials were not designed to target cancer patients bearing specific molecular subtypes. BET inhibitors are generally believed to preferentially suppress the transcription of super-enhancer-driven genes [38]. Super-enhancers were found at the driver oncogenes in many cancer types. It is noteworthy that an oncogene may be common to multiple cancer types, whereas the associated super-enhancers could be cancer type-specific. It follows that the super-enhancers may play an important role in determining the cancer cell-type-specific response to BET inhibitors. Indeed, it has been proposed that BET inhibitors may work preferentially in selective cancer subtypes. Therefore, it is important to elucidate the biology of the specific driver oncogenes (including *MYC* and BCL family genes) regulated by the Brd/BET proteins in individual cancer types. The information may allow the selection of a specific patient population to benefit from the BET inhibitors in clinical trials. 

Recent preclinical works have identified a few classes of chemotherapeutic drugs (including tyrosine kinase inhibitors, CDK inhibitors, proteasome inhibitors, and immunotherapies) for combination with BET inhibitors to enhance the therapeutic outcome and also to overcome resistance to BET inhibitors. It will be important to employ robust pharmacodynamic biomarkers to evaluate the efficacy of the novel drug combinations. Furthermore, early-generation BET inhibitors (notably, JQ1) have a short half-life and undesirable pharmacokinetic properties. High drug concentrations were usually used in preclinical studies for the evaluation of BET inhibitors. Detailed pharmacokinetics studies should be performed to establish the correlation between the dose, therapeutic efficacy and toxicity of the promising BET inhibitors during clinical investigation. 

## 10. Conclusions 

Extensive research efforts have been made to optimize BET inhibitors to unleash the full potential of this novel class of anticancer agents. The major obstacles hindering the clinical applications of BET inhibitors include the “class-effect” toxicities and the undesirable pharmacokinetic properties of the early-generation drug candidates. The design of the BD-selective and/or specific BET inhibitors has overcome some of the hurdles in their clinical application. Further research with an aim to achieve cancer cell-type- and organ site-specificity will allow the selection of a specific patient population who may benefit the most from this new class of drugs. 

## Figures and Tables

**Figure 1 molecules-28-03043-f001:**
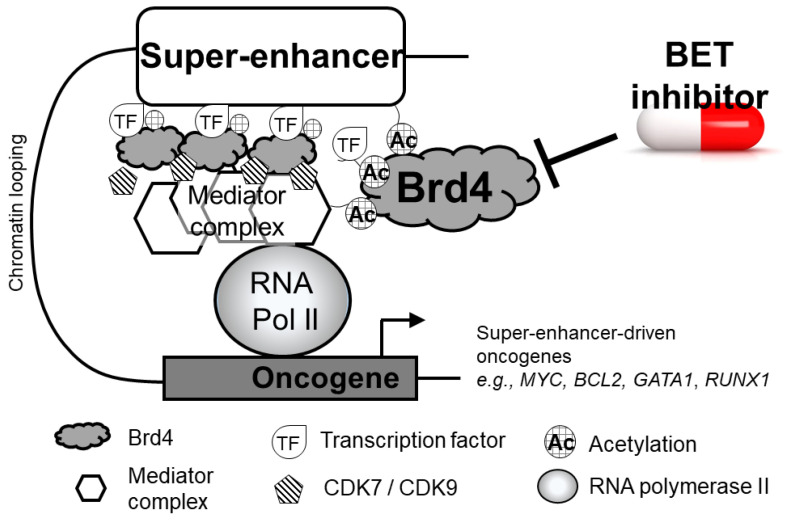
Therapeutic targeting of super-enhancers by BET inhibitors. Brd4 binds to heavily acetylated (Ac) lysines in super-enhancer complexes and transcription factors (TFs), thereby bringing them together and facilitating transcriptional activation and elongation via RNA polymerase II (RNA Pol II) and Mediator. BET inhibitors disrupt the interaction between Brd4 and acetylated lysine residues of super-enhancer complexes to suppress activation of oncogenes.

**Figure 2 molecules-28-03043-f002:**
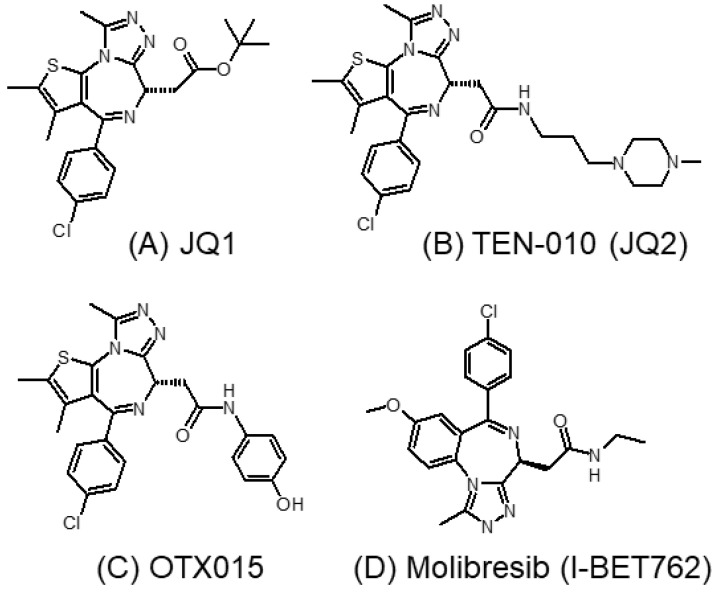
Chemical structures of the pan-BET inhibitors discussed: (**A**) JQ1, (**B**) JQ2, (**C**) OTX015, and (**D**) I-BET762.

**Figure 3 molecules-28-03043-f003:**
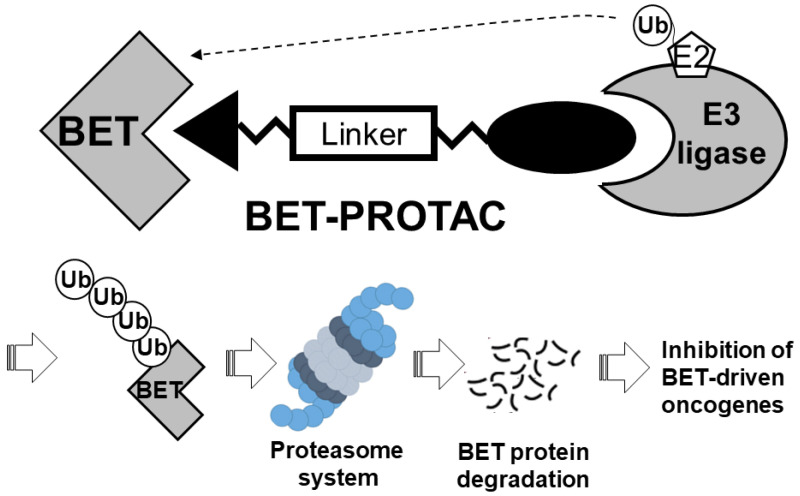
Schematic diagram showing the general design of a BET-protein-targeting proteolysis-targeting chimera (PROTAC). A typical PROTAC consists of a BET ligand connected to an E3 ligand via a linker. BET protein is brought to close proximity to the E3 ligase in the presence of the PROTAC, which is subsequently ubiquitinated (Ub) and subjected to rapid degradation.

**Figure 4 molecules-28-03043-f004:**
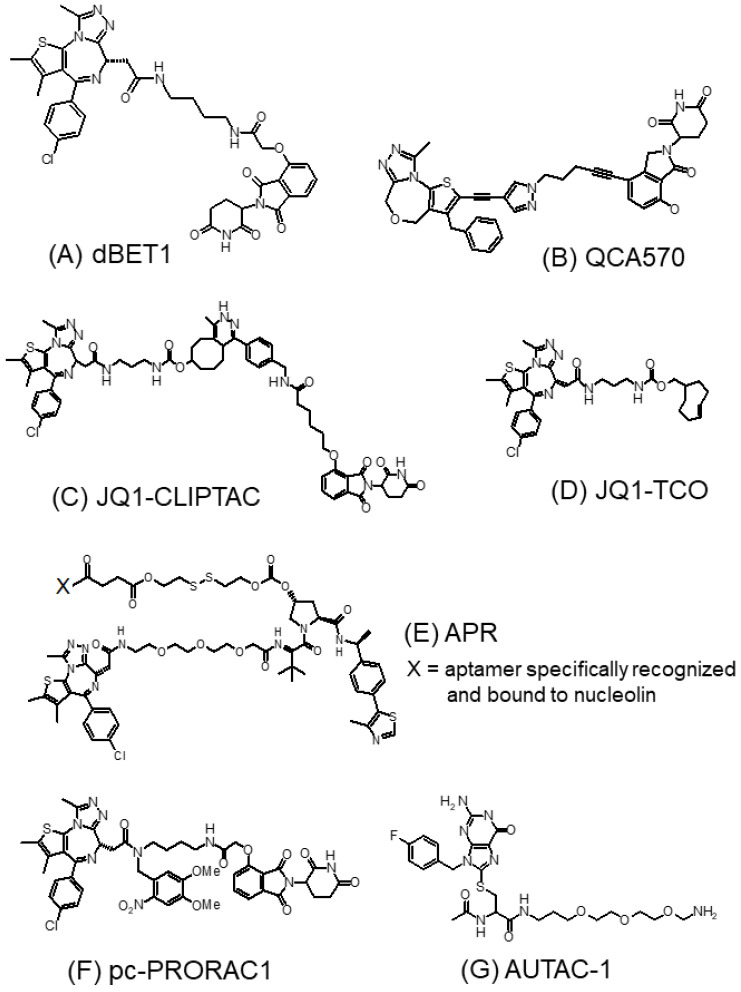
Chemical structures of the BET-PROTACs discussed: (**A**) dBET1, (**B**) QCA570, (**C**) JQ1-CLIPTAC, (**D**) JQ1-TCO, (**E**) APR, (**F**) pc-PRORAC1, and (**G**) AUTAC-1.

**Figure 5 molecules-28-03043-f005:**
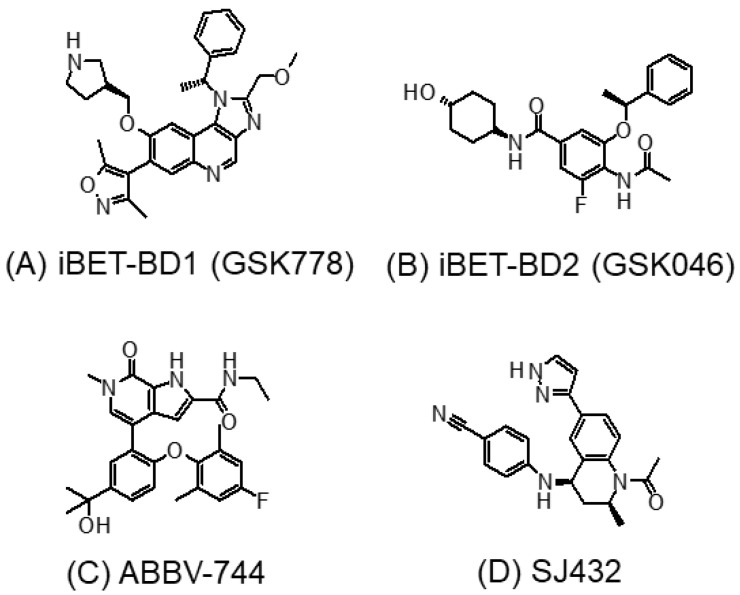
Chemical structures of the BD1/BD2-selective BET inhibitors discussed: (**A**) GSK778, (**B**) GSK046, (**C**) ABBV-744, and (**D**) SJ432.

**Figure 6 molecules-28-03043-f006:**
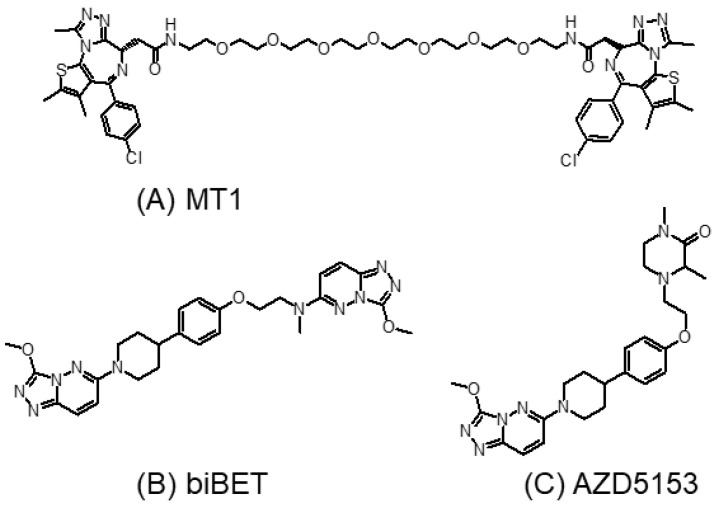
Chemical structures of the bivalent BET inhibitors discussed: (**A**) MT1, (**B**) biBET, and (**C**) AZD5153.

**Figure 7 molecules-28-03043-f007:**
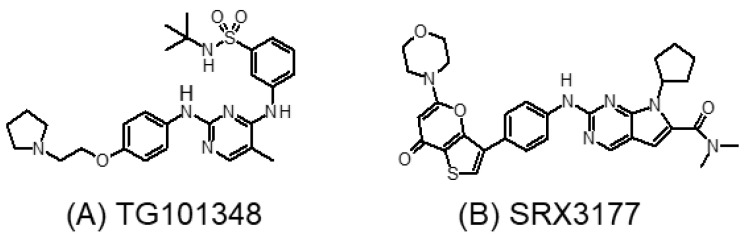
Chemical structures of the tyrosine kinases and BET dual inhibitors discussed: (**A**) TG101348, and (**B**) SRX3177.

**Figure 8 molecules-28-03043-f008:**
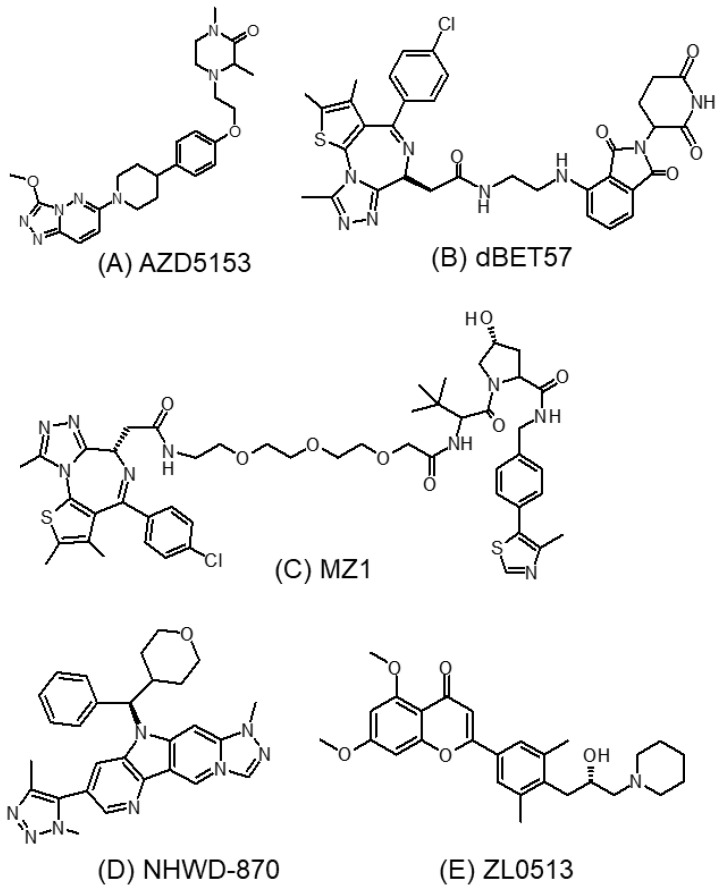
Chemical structures of the Brd4-selecitve BET inhibitors discussed: (**A**) AZD5153, (**B**) dBET57, (**C**) MZ1, (**D**) NHWD-870, and (**E**) ZL0513.

**Figure 9 molecules-28-03043-f009:**
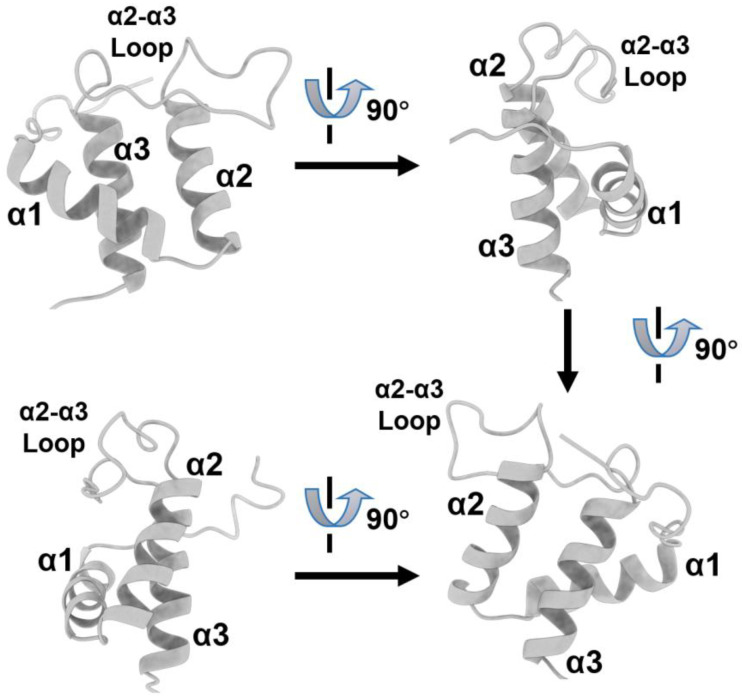
The annotations of secondary structures in the Brd4-ET domain. Alpha helixes (α) are indicated. The model is from the solution NMR structure of the Brd4-ET bound with MLV-IN EBM (PDB code: 2N3K).

**Figure 10 molecules-28-03043-f010:**
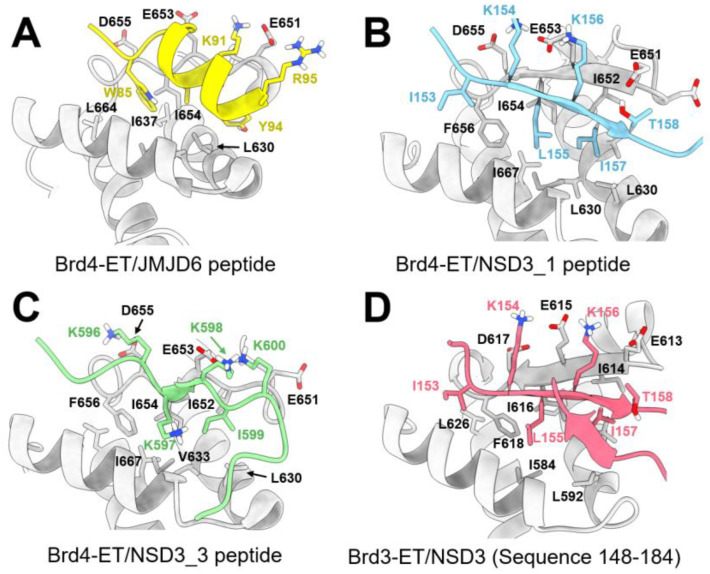
The scheme of the Brd3- or Brd4-ET domain interacting with NSD3 (blue, green, and pink) and JMJD6 (yellow) peptides. Interacting residues on ET are indicated in black, and interacting residues on corresponding peptides are colored as indicated. ET domains are rotated to present the best angle to see the interacting residues. The PDB codes of associated structures are (**A**) 6BNH, (**B**) 2NCZ, (**C**) 2ND1, and (**D**) 7JYN.

**Figure 11 molecules-28-03043-f011:**
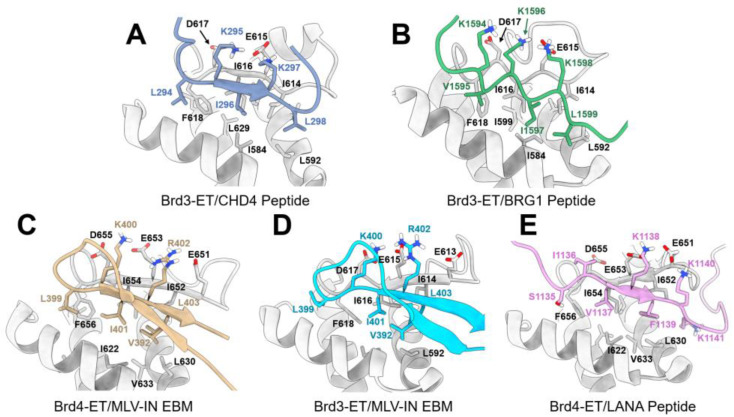
The scheme of the Brd3- or Brd4-ET domain interacting with murine leukemia virus (MLV) integrase (IN) ET-binding motif (MLV-IN EBM) (peach and cyan), LANA (pink), CHD4 (dark blue), and BRG1 (green) peptides. Interacting residues on ET are indicated in black, and interacting residues on corresponding peptides are colored as indicated. ET domains are rotated to present the best angle to see the interacting residues. The PDB codes of associated structures are (**A**) 6BGG, (**B**) 6BGH, (**C**) 2N3K, (**D**) 2ND0, and (**E**) 7JYZ.

**Table 1 molecules-28-03043-t001:** The eight subfamilies of bromodomain (BRD)-containing proteins encoded in the human genome.

Subfamily Number	Representative Members	Cellular Function(s)
I	PCAF	Histone acetyltransferase
	GCN5L	Histone acetyltransferase
	FALZ	Chromatin remodeling factor
	CECR2	Chromatin remodeling factor
II (the BET BRDs)	BRD2	Transcriptional regulator
	BRD3	Transcriptional regulator
	BRD4	Transcriptional regulator
	BRDT	Chromatin remodeling factor
III	BRD8B	Transcriptional regulator
	CREBBP	Histone acetyltransferase
	EP300	Histone acetyltransferase
	BAZ1B	Tyrosine-protein kinase; transcriptional regulator
	BRWD3 domain 2	JAK/STAT signaling
	PHIP domain 2	Insulin signaling
IV	BRD7	Transcriptional regulator
	BDR1	Transcriptional regulator
	BDPF1	Transcriptional regulator
	ATAD2	Transcriptional regulator
	BRD9	Unknown
	BRPF3	Transcriptional regulator
V	TRIM66	Transcriptional repressor
	TRIM33	Ubiquitin E3 ligase; transcriptional regulator
	TIF1α	Ubiquitin E3 ligase; transcriptional regulator
	SP100	Transcriptional regulator
	SP110	Transcriptional regulator
VI	MLL	Histone methyltransferase
	TRIM28	SUMO E3 ligase; transcriptional regulator
VII	TAF1	Transcriptional initiation
	TAF1L	Transcriptional initiation
	BRWD3 domain 1	JAK/STAT signaling
	PHIP domain 1	Insulin signaling
VIII	ASH1L	Histone-lysine methyltransferase
	SMARCA2A	Chromatin remodeling factor
	SMARCA4	Chromatin remodeling factor
	PB1	Transcriptional regulator

**Table 2 molecules-28-03043-t002:** Representative clinical trials investigating novel BET inhibitors for treatment of hematological and solid tumors in recent years.

Targeting Isoform	BET Inhibitor	Study Phase/ Patient Population	ClinicalTrials.gov Numbera (Current Status)
pan-BET	ABBV-075 (Mivebresib)	Phase I/1st-in-human, dose escalation study/ Breast, NSCLC, AML, MM, prostate, SCLC, non-Hodgkin lymphoma	NCT02391480 (completed; findings reported in [205])
	EP31670 (dual BET and CBP/p300 inhibitor)	Phase I/dose escalation study/ Castration-resistant prostate cancer and NUT carcinoma	NCT05488548 (not yet recruiting)
	FT-1101	Phase I/Ib/Safety, pharmacokinetics and pharmacodynamics study/ AML, myelodysplastic syndrome, non-Hodgkin lymphoma	NCT02543879 (completed; findings reported in [127])
	GS-5829 (Alobresib)	Phase Ib/II/ Part I—Safety, tolerability, pharmacokinetics and pharmacodynamics study of GS-5829 as a single agent Part II—Combination of GS-5829 and enzalutamide in patients with metastatic castration-resistant prostate cancer	NCT02607228 (completed; findings reported in [206])
	GSK525762 (Molibresib)	Phase I/II Safety, pharmacokinetics and pharmacodynamics study/ Patients with relapsed, refractory hematologic malignancies	NCT01943851 (completed; findings reported in [58])
	INCB054329 (BD2-selective)	Phase I/II Dose escalation, safety and tolerability study Solid tumors and hematological malignancy	NCT02431260 (terminated by sponsor in Jan 2018 due to pharmacokinetics variability)
	RO680810	Phase I/dose escalation study/ NUT carcinoma, advanced solid tumors, or DLBCL	NCT01987362 (completed; findings reported in [207])
	TEN-010	Phase I/dose escalation study/ AML and myelodysplastic syndrome	NCT02308761 (completed; findings reported in [53])
	TEN-010	Phase I/Part 1—dose escalation; Part 2—Expansion cohort in patients with selected malignancies/ Advanced solid tumors	NCT01987362 (completed; findings reported in [207])
	ZEN003694	Phase I/Safety and tolerability study/ Metastatic CRPC	NCT02705469 (completed)
	ZEN003694	Phase II/Single group assignment Squamous cell lung cancer patients harboring NSD3 amplification	NCT05607108 (recruiting)
Brd2/3/4	ABBV-744 (BD2-selective)	Phase I/Safety and pharmacokinetics study/ Breast, NSCLC, AML, MM, prostate, SCLC, non-Hodgkin lymphoma	NCT02391480 (completed; findings reported in [208])
	I-BET151 (GSK2820151)	Phase I/dose escalation study/ Advanced or recurrent solid tumors	NCT02630251 (terminated in 2017 due to development of another BET inhibitor (GSK525762) with a better understanding of the risk–benefit profile
	I-BET762 (GSK525762; Molibresib)	Phase I/II/dose escalation study/ Relapsed or refractory hematologic malignancies	NCT01943851 (completed; findings reported in [209])
	I-BET762 (GSK525762; Molibresib)	Phase I/Open label cross-over study to evaluate the effect of itraconazole and rifampicin on the pharmacokinetics of I-BET0762)	NCT02706535 (completed; findings reported in [210])
	OTX-015 (Birabresib)	Phase I/dose finding study/ AML, DLBCL, ALL, MM	NCT01713582 (completed; findings reported in [165])
	OTX-015 (Birabresib)	Phase IIa/dose optimization/ Recurrent GBM	NCT02296476 (terminated due to lack of clinical activity and not due to safety reasons)
	OTX-015 (Birabresib)	Phase IB/dose exploration trial/ NMC, TNBC, NSCLC, CRPC	NCT02698176 (terminated due to limited efficacy and not due to safety reasons)
	OTX-015 (Birabresib)	Phase IB/ NMC, TNBC, NSCLC with rearranged ALK gene/fusion protein or KRAS mutation, CRPC, PDAC	NCT02259114 (completed; findings reported in [211])
Brd2/4	CC-90010 (Trotabresib)	Phase I/1st-in-human/dose escalation and expansion/ Advanced solid tumors and relapsed/refractory non-Hodgkin lymphoma	NCT03220347 (completed; findings reported in [94])
Brd4	AZD5153	Phase I/dose escalation study/ Relapsed or refractory malignant solid tumors, lymphomas	NCT03205176 (completed)
	AZD5153	Phase I/Platform protocol for the treatment of relapsed/refractory/aggressive non-Hodgkin lymphoma	NCT03527147 (completed)
	CPI-0610 (Pelabresib)	Phase I/1st-in-human Patients with relapsed/refractory lymphomas	NCT01949883 (completed; findings reported in [212])
	CPI-0610	Phase I/ Patients with previously treated multiple myeloma	NCT02157636 (completed)
	CPI-0610	Phase 1 (dose escalation of CPI-0610 in patients with hematological malignancies) Phase 2 (dose expansion of CPI-0610 with and without ruxolitinib in patients with myelofibrosis and essential thrombocytopenia)	NCT02158858 (active, not recruiting)
	NUV-868 (BD2-selective)	Phase 1/1st-in-human dose escalation and expansion study/ Advanced solid tumors Phase 2/combination with olaparib or enzalutamide/ Advanced solid tumors	NCT05252390 (recruiting)
	PLX51107	Phase I/combination of PLX51107 and azacytidine/ Myelodysplastic syndrome and AML	NCT04022785 (completed in October 2022)

Abbreviations: ALL = acute lymphoblastic leukemia; AML = acute myeloid leukemia; CRPC = castration-resistant prostate cancer; DLBCL = diffuse large B-cell lymphoma; GBM = glioblastoma multiforme; MM = multiple myeloma; NMC = NUT midline carcinoma; NSCLC = non-small-cell lung cancer; PDAC = pancreatic ductal adenocarcinoma; TNBC = triple-negative breast cancer.

## Data Availability

Not applicable.

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
