# Peer review of "BET Bromodomain Inhibitors: Novel Design Strategies and Therapeutic Applications"

_molecules, 2023, doi:10.3390/molecules28073043_

Round 1

Reviewer 1 Report

The authors make a useful compilation of the various small molecule-based BET inhibitors. However, the inclusion of the chemical structures of the molecules collected in the review is missing. The inclusion of these structures is vital to understand in depth the evolution of this strategy related to these inhibitors and it is necessary to include them in the paper.

Author Response

Thank you so much for the favorable comment from the reviewer.

As recommended by the reviewer, chemical structures of the molecules discussed in the manuscript have been included (Figures 2,4,5,6,7,8).

Reviewer 2 Report

Manuscript molecules-2292112, is very well written review on the last advancements in the development of BET inhibitors. In my opinion the manuscript can be accepted for publication after the following modifications:

- Figures with the chemical structures of all the discussed molecules have to be included;

-subparagraphs numbering of pan-BET inhibitors should be revised. Moreover, in my opinion. BET-PROTACs, being unselective, should be included in the paragraph 3.1 on pan-BET inhibitors.

Author Response

Thank you so much for the favorable comments from the reviewer!

As recommended by the reviewer, the following revisions have been made to the manuscript:

(1) Chemical structures of the molecules discussed in the manuscript are now included (Figure 2,4,5,6,7,8)

(2) Subparagraph numbering of pan-BET inhibitors is revised

(3) "BET-PROTACs" are moved to paragraph3.1 "pan-BET inhibitors"

Reviewer 3 Report

molecules-2292112

This is a well written, interesting and informative review on the research for the development of small molecule inhibitors of BET proteins. There are not many things to be altered in this work and in my opinion it deserves to be published.

I have a main suggestion to propose: From my point of view, it would be much preferable if the authors could include the chemical structures of small molecule inhibitors in the review. This would greatly facilitate medicinal chemists to understand the structural requirements for a compound to act as an inhibitor.

A minor point could be to add appropriate reference(s) in the 1st paragraph of section 2.2 (pages 3-4).

Author Response

Thank you so much for the favorable comment from the reviewer and for the valuable suggestions. The manuscript has been revised accordingly.

  • Chemical structures of the molecules discussed in the manuscript have been included (Figures 2,4,5,6,7,8).
  • Appropriate references have been added to the 1st paragraph of section 2.2 (reference # 6,17, 18-21, 22,23).